



# Retrieving Tropospheric Refractivity Structure using Interferometry of Aircraft Radio Transmissions

Ollie Lewis[1], Chris Brunt[1], Malcolm Kitchen[1], Neill E. Bowler[2], and Edmund K. Stone[2]

[1]Department of Physics and Astronomy, University of Exeter, Exeter, EX4 4QL, United Kingdom
[2]Met Office, Fitzroy Road, Exeter, EX1 3PB, United Kingdom

**Correspondence:** Ollie Lewis (osl202@exeter.ac.uk)

**Abstract.** Detailed measurements of atmospheric humidity in the lower atmosphere are currently difficult and expensive to obtain. For this reason, there is interest in the development of low-cost, high-volume opportunistic technologies to acquire measurements of tropospheric humidity. We demonstrate the use of interferometry to measure the atmospheric refraction of the Automatic Dependent Surveillance-Broadcast (ADS-B) radio transmission routinely broadcast by commercial aircraft.

Atmospheric refraction is strongly influenced by changes in humidity, and refractivity observations have proved to be an effective source of humidity information for numerical weather prediction models. A prototype ADS-B interferometer has been developed that can simultaneously perform angle-of-arrival (AoA) interferometry and decode ADS-B signals. Combining the measured AoA of the ADS-B signal with the known position of the aircraft (information contained within the ADS-B signal) allows the bending of the signal due to refraction to be determined. Combining the measured bending of numerous ADS-B

signals allows for information concerning the refractivity structure to be extracted. An adjoint model was derived and used to retrieve synthetic one-dimensional refractivity profiles in a variety of atmospheric conditions. The results from an experiment using a prototype ADS-B interferometer are shown and initial refractivity profiles are retrieved. Sources of uncertainty in the observations and the retrieved refractivity profiles were explored and future work was suggested.

## 1 Introduction

The global distribution of water vapour partially governs the transport of heat and the radiation budget of the atmosphere

(Trenberth et al., 2009; Held and Soden, 2000). In the lower atmosphere, the spatial and temporal variability in water vapour strongly influences the formation and intensity of extreme weather events through condensation and subsequent release of latent heat (Trenberth et al., 2005). The high sensitivity of atmospheric processes to water vapour presents the requirement for accurate observations for both climatological and meteorological purposes. Despite its importance in climate and driving





extreme weather events, the spatial and temporal variability of tropospheric humidity is difficult to capture adequately using
existing observing systems (Rieckh et al., 2018). Radiosondes provide the principal source of in situ tropospheric water vapour
observations, capable of retrieving high resolution vertical profiles of humidity. However, the limited spatial distribution of
launch sites and low frequency of radiosonde ascents (typically once daily) substantially restrict the ability to retrieve the
full four-dimensional structure of water vapour in the lower atmosphere. Another source of in-situ humidity observations is
provided by the Aircraft Meteorological Data Relay (AMDAR). The onboard water vapour sensing system (WVSS) instrument
can provide detailed profiles of humidity, particularly as the aircraft traverses the moist lower atmosphere (Petersen et al.,
2016). However, only a fraction of AMDAR reports currently contain usable humidity data, with the majority restricted to
North America (Petersen et al., 2016; Ingleby et al., 2019; Pauley and Ingleby, 2022).

Developments in remote sensing of atmospheric water vapour have substantially improved the ability of observing systems
to capture the variability and distribution of tropospheric humidity. Concerning remote sensing of humidity, surface-based re-
trievals using lidars can provide excellent temporal coverage of the variability in humidity throughout the extent of the lower
atmosphere directly above the sensor (Guo et al., 2023). However, individual units are expensive and limited in number, re-
stricting the distribution of retrieved vertical profiles of tropospheric humidity for use in numerical weather prediction (NWP)
(Gaffard et al., 2021). The maturation of satellite-based remote sensing of humidity has dramatically increased the global cov-
erage of humidity observations. Satellite observations, such as those from passive microwave and infrared-sounding satellites,
have proved to be a crucial source of indirect humidity observations. In particular, microwave humidity-sounding has had a
substantial positive impact on short-range forecasting skill (Geer et al., 2017). Microwave humidity-sounding satellite obser-
vations are an especially crucial source of humidity information over the data-sparse open oceans, where the positive impact of
radiance-derived humidity observations in these regions has been demonstrated to propagate downstream with the atmospheric
flow over much of Europe and landmasses in the southern hemisphere. However, the large weighting functions (2-3 km) used to
quantify the vertical distribution of humidity severely limit the resolution of retrieved profiles (Rieckh et al., 2018; Wang et al.,
2024) relative to the vertical resolution of NWP models such as the Met Office United Kingdom (UK) Variable-resolution
(UKV) model (average vertical spacing in the troposphere is on the order $\sim 100$ m) (Tang et al., 2013). A valuable source
of indirect humidity information for use in NWP is provided through observations of refractivity. An example would be the
Global Navigation Satellite System (GNSS) radio occultation technique (Kuo et al., 2000; Healy and Thépaut, 2006), which
exploits the refractive bending of radio signals between a low Earth orbit (LEO) satellite and a GNSS satellite to retrieve the
refractive index (refractivity) structure of the atmosphere. Although GNSS radio occultation can provide accurate high reso-
lution vertical profiles of refractivity, the retrievals can be significantly impacted by strong gradients in refractivity (Xie et al.,
2006). Refractivity observations have also been obtained using radar phase shifts (Fabry et al., 1997; Weckwerth et al., 2005)
and zenith tropospheric delay (ZTD) measurements of GNSS satellite radio signals (Rohm, 2012; Zhao et al., 2019). However,
the enormous spatial and temporal variability in refractivity make accurate, high resolution retrievals challenging.

There is an increasing interest to acquire atmospheric observations using commercial aircraft, which are mandated to broad-
cast digitally encoded three-dimensional position via the Automatic Dependent Surveillance-Broadcast (ADS-B) system for
air traffic control and safety purposes (ICAO, 2012). Meteorological parameters derived using ADS-B and Mode-Selective



Enhanced Surveillance (Mode-S EHS) data include temperature (de Haan, 2011; Stone and Kitchen, 2015; Mirza et al., 2019)
and wind (de Haan et al., 2013; Mirza et al., 2016) data. Aircraft-derived observations have been exploited by the UK Met
Office (Stone and Pearce, 2016; Stone, 2018; Li, 2021) and Royal Netherlands Meteorological Institute (KNMI) (de Haan and
Stoffelen, 2012; de Haan et al., 2013) to retrieve high density wind data. The potential to retrieve atmospheric refractivity data
using aircraft-derived observations has recently been explored (Lewis et al., 2023a). A two-element interferometer was used to
measure the bending of refracted ADS-B signals and analogies with GNSS radio occultation were investigated (Lewis et al.,
2023b). In this study we demonstrate the potential to retrieve one-dimensional (1D) refractivity profiles using synthetic and
real observations obtained using a prototype ADS-B interferometer.

## 2   Methods

### 2.1   Interferometry of radio transmissions

Atmospheric refraction of ADS-B radio signals transmitted by aircraft can be detected using vertically-orientated, two-element
interferometers. The total bending of the radio signal can be determined by combining the interferometrically observed incident
angle-of-arrival (AoA) with the known point of origin of the signals (information encoded within the ADS-B transmission).
The incident AoA measurement technique is described in detail in Lewis et al. (2023a). Measurable bending only occurs at
grazing incidence (AoA $\lesssim 2°$), therefore a clear view of the horizon is required to acquire meteorologically useful observations.
A measurable phase difference between the signal received at the upper antenna and the signal received at the lower antenna
is induced by the path length excess (geometry depicted in Fig. 1). The phase difference, $\phi$, can be written in terms of the path
excess as

$$\phi = 2\pi \frac{\boldsymbol{B} \cdot \hat{\boldsymbol{z}}}{\lambda} = 2\pi \frac{B\sin(\beta)}{\lambda}, \tag{1}$$

where $\boldsymbol{B}$ is the vertical baseline vector (displacement between the upper and lower antennas), $\hat{\boldsymbol{z}}$ is the unit vector in the
direction of the received ADS-B signal, $\beta$ is the incident AoA and $\lambda$ is the signal wavelength ($\sim 0.275$ m for a 1090 MHz
transmission). An Ettus X300 software-defined radio (SDR) was used to determine the phase difference by computing the
conjugate product of the signals from each interferometer element.

### 2.2   Atmospheric refraction

The refractive properties of the atmosphere can be described using refractivity, which is the parts-per-million (ppm) excess of
the refractive index above the unity vacuum value. The total refractivity at a given point in the atmosphere can be decomposed
into the dry refractivity (contribution of the dry atmospheric gases to the total refractivity, $N_{\text{dry}}$) and the wet refractivity
(contribution of water vapour to the total refractivity, $N_{\text{wet}}$) as




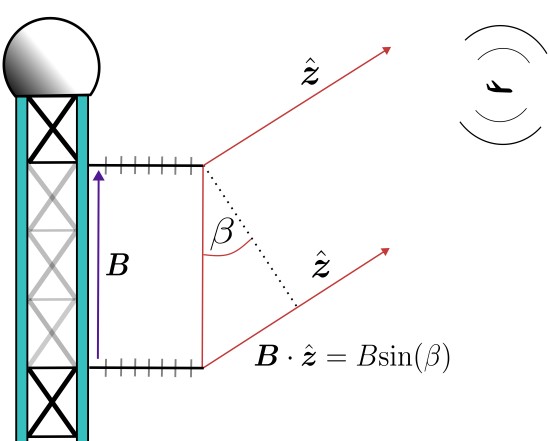

**Figure 1.** Geometry of the ADS-B interferometry observing system.

$$N_{\text{dry}} = q_1 \frac{P}{T}, \tag{2}$$

$$N_{\text{wet}} = q_2 \frac{e}{T^2}, \tag{3}$$

where $q_1$ and $q_2$ are empirical constants given by $q_1 = 77.6$ K hPa$^{-1}$ and $q_2 = 3.73 \times 10^5$ K$^2$ hPa$^{-1}$ respectively (Smith and
Weintraub, 1953). The air pressure, temperature and partial pressure of water vapour are given by $P$ [hPa], $T$ [K] and $e$ [hPa]
respectively. Other more accurate formulae exist for the calculation of refractivity (e.g. Aparicio and Laroche (2011)) but the
simple form is still commonly used. The contribution of water vapour to the total refractivity is generally around 20% near the
surface (Möller, 2017) but can fluctuate rapidly over short spatial and temporal scales. Refractivity data was obtained using the
12:00 Coordinated Universal Time (UTC) launch radiosonde data from Watnall, Nottinghamshire (UK) (World Meteorological
Organisation (WMO) station number 03354) on the 22 September 2023. The radiosonde data used in this study are available
from the University of Wyoming, United States of America. Figure 2 shows the tropospheric dry refractivity profile, as well
as the modulation due to the wet refractivity contribution. Above an altitude of around 7 km, the contribution of the wet
refractivity to the total refractivity is negligible.

To model the propagation of a radio signal through an atmosphere with a radially-directed refractive index gradient, we
apply ray-tracing techniques used in geometric optics. The ray path in a spherically symmetric atmosphere can be defined
using Snell's law

$$n \cdot (a+h) \cdot \cos(\epsilon) = \text{constant}, \tag{4}$$





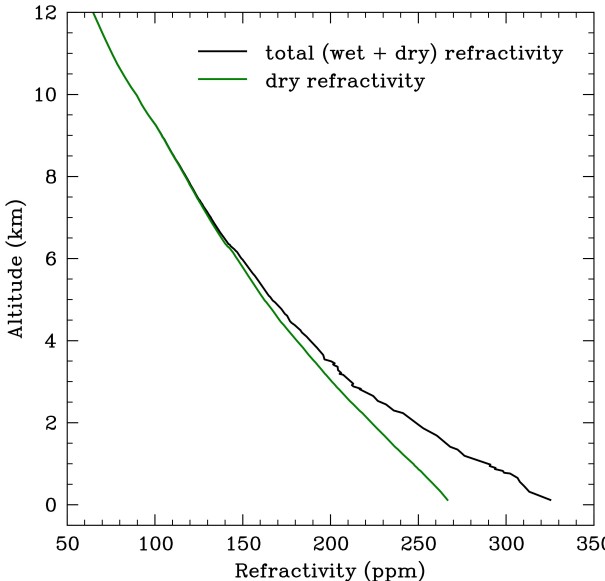

**Figure 2.** The total refractivity (solid black line) and dry refractivity (solid green line) versus altitude (in kilometres) retrieved using the 12:00 UTC launch radiosonde data from Watnall on the 22 September 2023.

where $n$ is the refractive index, $h$ is the height of the ray above the surface, $\epsilon$ is the local elevation angle of the ray and $a$ is the radius of the Earth. The ray propagation can be formulated as an initial value problem, dependent on the initial position and direction of the ray. Here we use the second-order ordinary differential equation (SODE) derived by Zeng et al. (2014) to simulate the propagation. The SODE model is derived using the initial assumption that the height, $h$, is dependent on the along-beam range, $r$. Therefore

$$n[h(r)] \cdot (a + h(r)) \cdot \cos(\epsilon) = \text{constant}. \tag{5}$$

Differentiating (5) with respect to $r$ allows the second-order differential equation to be derived (the full derivation is described in detail in Zeng et al. (2014))

$$\frac{\mathrm{d}^2 h}{\mathrm{d}r^2} = \left(1 - \left(\frac{\mathrm{d}h}{\mathrm{d}r}\right)^2\right)\left(\frac{1}{n}\frac{\mathrm{d}n}{\mathrm{d}h} + \frac{1}{a+h}\right). \tag{6}$$

The introduction of the variable $u$ allows the SODE to be separated into a system of first-order differential equations

$$\frac{\mathrm{d}h}{\mathrm{d}r} = u, \tag{7}$$





$$\frac{\mathrm{d}u}{\mathrm{d}r} = (1 - u^2)\left(\frac{1}{n}\frac{\mathrm{d}n}{\mathrm{d}h} + \frac{1}{a+h}\right), \tag{8}$$

which can be integrated over the domain $[0, r_{\mathrm{f}}]$ (where $r_{\mathrm{f}}$ defines the endpoint of the ray) to recover the ray path. We choose to integrate equations (7) and (8) using a third-order Runge-Kutta scheme. The distance travelled by the ray at each step, $i$, along the surface of the Earth, $s$, is given by

$$s_{i+1} = s_i + \sin^{-1}\left(\frac{\cos(\epsilon_i)\Delta r}{a + h_{i+1}}\right)a, \tag{9}$$

where $\epsilon_i = \sin^{-1}(u_i)$ is the local elevation angle of the ray at each step. The initial height and direction of the ray are required to model the refraction of a radio signal through the atmosphere. The altitude of the aircraft can be determined, however, the initial emission angle of the radio signal is unknown. The refraction is therefore modelled in a time-reversed frame from the position of the receiver to the reporting aircraft. A prior (first-guess) refractivity profile is used. The squared difference between the altitude of the model ray after termination and the reported height of the aircraft is used to compute a penalty function. The best estimate of the true refractivity profile is determined by minimising the penalty. We explore this optimisation problem in more detail in Section 2.3.

### 2.3 Optimisation of the refractive index profile

The optimisation problem can be formulated as

$$\min_n \quad J = \sum_{i=0}^{N_{\mathrm{B}}} j_i = \sum_{i=0}^{N_{\mathrm{B}}} ((h_{\mathrm{f}})_i - (h_{\mathrm{t}})_i)^2, \tag{10a}$$

$$\text{subject to} \quad \dot{h}(r; n, h_0, u_0) - u = 0, \qquad \forall r \in [0, r_{\mathrm{f}}], \tag{10b}$$

$$\dot{u}(r; n, h_0, u_0) - (1 - u^2)\left(\frac{1}{n}\frac{\mathrm{d}n}{\mathrm{d}h} + \frac{1}{a+h}\right) = 0, \qquad \forall r \in [0, r_{\mathrm{f}}], \tag{10c}$$

$$h(0; n, h_0, u_0) = h_0, \tag{10d}$$

$$u(0; n, h_0, u_0) = u_0, \tag{10e}$$

where $h$ and $u$ describe the height (position) and (sine) direction of the ray respectively and $\dot{h}$ and $\dot{u}$ are their respective derivatives with respect to $r$. The initial position and direction of the ray are given by $h_0$ and $\beta$ respectively. $J$ is the total penalty function describing the sum of the individual penalty functions, $j_i$, corresponding to the squared difference between the end height of the $i^{\mathrm{th}}$ ray, $(h_{\mathrm{f}})_i$, and the $i^{\mathrm{th}}$ target height, $(h_{\mathrm{t}})_i$ ($i = 0, 1, 2, \ldots, N_{\mathrm{B}}$, where $N_{\mathrm{B}}$ is the number of broadcasts used in the optimisation problem). In the case of ADS-B interferometry, the target position will be the reported position of the aircraft.



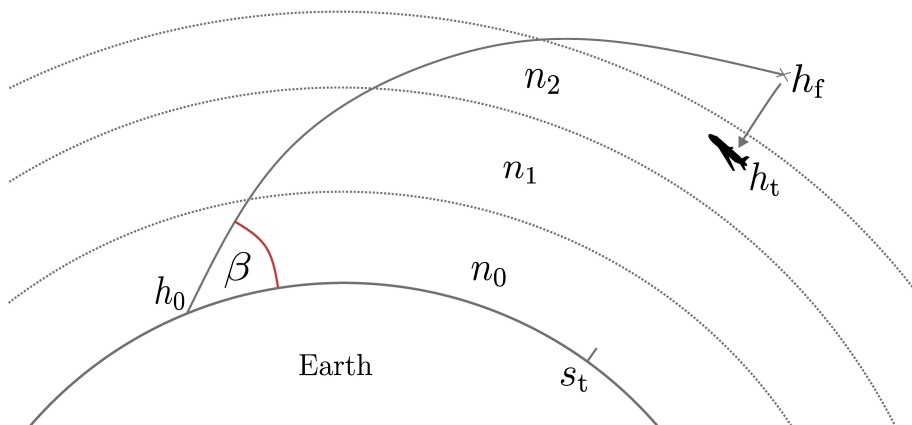

**Figure 3.** Illustration of the single-ray optimisation problem. A ray with initial position, $h_0$, and elevation (exaggerated in the diagram for illustrative purposes), $\beta$, is traced through an initial-guess refractive index profile until it reaches the aircraft distance, $s_t$. The penalty function for a single ray is calculated as the squared difference between the ray endpoint, $h_f$, and the target (aircraft) position, $h_t$.

Figure 3 shows the geometry of the system to be optimised for a single ray. The refractive index profile, $\boldsymbol{n}$, was optimised such that a ray traced through the profile with an initial position and direction given by $h_0$ and $\beta$ respectively will terminate at the target (aircraft) position, $h_t$. The gradient of the penalty function, $J$, with respect to the refractive index profile, $\boldsymbol{n}$, was required to minimise $J$ via gradient descent. The gradient of the individual penalty function for a single ray can be approximated by computing finite differences over the elements of the refractive index profile $\boldsymbol{n} \in \mathbb{R}^M$ (where $M$ is the number of refractive index values along the ray trajectory) as

$$\frac{\mathrm{d}j[h_f(\boldsymbol{n})]}{\mathrm{d}n_k} \approx \frac{j[h_f(\boldsymbol{n} + \delta n \boldsymbol{e}_k)] - j[h_f(\boldsymbol{n} - \delta n \boldsymbol{e}_k)]}{2\delta n_k}, \tag{11}$$

where $n_k$ is the $k^{\text{th}}$ element of the modified refractive index profile. The $k^{\text{th}}$ element of the vector $\boldsymbol{e}_k$ ($\boldsymbol{e}_k \in \mathbb{R}^M$) is unity and zero otherwise and $\delta n$ is a small increment to the $k^{\text{th}}$ element of $\boldsymbol{n}$. Each gradient evaluation with respect to a single element of the modified refractive index profile requires integrating equations (10b) and (10c) over the entire ray trajectory to determine the change in the end position of the ray, $h_f$. The penalty gradient calculation using finite differences for a single ray is expensive and quickly becomes impractical when evaluating the total penalty function gradient for $N_B$ rays when $N_B$ is large. In a previous investigation by Lewis et al. (2023a), the total penalty function given by (10a) was minimised using simulated annealing, a stochastic global search optimisation algorithm (Van Laarhoven et al., 1987). Although simulated annealing is an effective method for optimising high-dimensional problems with potentially many local minima, the algorithm is very computationally expensive and requires thousands of iterations to converge on a good solution. For this reason, the optimisation procedure using the adjoint state method (Zou et al., 1997; Gill et al., 1981) was used to reduce the computational cost of retrieving





the refractivity profile. Applying the adjoint state method to our original optimisation problem, the constrained optimisation problem described by (10) can be converted into an unconstrained optimisation problem using Lagrange multipliers

$$\mathcal{L} = j - \int_0^{r_\mathrm{f}} \psi\left(\dot{h} - u\right) \mathrm{d}r - \int_0^{r_\mathrm{f}} \mu\left(\dot{u} - (1-u^2)\left(\frac{1}{n}\frac{\mathrm{d}n}{\mathrm{d}h} + \frac{1}{a+h}\right)\right)\mathrm{d}r, \tag{12}$$

where $\psi$ and $\mu$ are the adjoint variables associated with the state variables $h$ and $u$ respectively. The integration of equations (10b) and (10c) over the domain $[0, r_\mathrm{f}]$ (where $r_\mathrm{f}$ parameterises the endpoint of the ray) defines the trajectory of the ray. The minimisation of (12) corresponds to finding the stationary points of the Lagrangian with respect to the trajectory defined by $h$ and $u$. Therefore any variations in the Lagrangian with respect to the trajectory are equal to zero (that is, $\delta\mathcal{L}|_h = \delta\mathcal{L}|_u = 0$). The adjoint variables enforce this constraint. Solving for the stationary points of the Lagrangian with respect to $h$ and $u$ we

find

$$\psi|_{r=r_\mathrm{f}} = \frac{\partial j}{\partial h}, \tag{13}$$

$$\mu|_{r=r_\mathrm{f}} = \frac{\partial j}{\partial u}, \tag{14}$$

$$\frac{\mathrm{d}\psi}{\mathrm{d}r} = \mu(1-u^2)\left(\left(\frac{1}{n}\frac{\mathrm{d}n}{\mathrm{d}h}\right)^2 - \frac{1}{n}\frac{\mathrm{d}^2n}{\mathrm{d}h^2} + \frac{1}{(a+h)^2}\right), \tag{15}$$

$$\frac{\mathrm{d}\mu}{\mathrm{d}r} = 2\mu u\left(\frac{1}{n}\frac{\mathrm{d}n}{\mathrm{d}h} + \frac{1}{a+h}\right) - \psi. \tag{16}$$

The full derivation is described in Appendix A. The initial conditions of the adjoint variables $\psi$ and $\mu$ are defined at the endpoint of the ray trajectory, $h|_{r=r_\mathrm{f}}$. The adjoint equations are therefore evolved in reverse from the endpoint to the initial position of the ray. In the simulated retrievals, an atmosphere with spherical symmetry was assumed and values were computed using a linear interpolation scheme, hence $\mathrm{d}^2n/\mathrm{d}h^2 = 0$. Since the refractive index generally decreases exponentially with height, the interpolation was performed using the natural logarithm of the refractive index values. Expressing the height-dependent natural

logarithm of the refractive index as $\tilde{n}(h) = \ln(n(h))$, equations (15) and (16) can then be written as

$$\frac{\mathrm{d}\psi}{\mathrm{d}r} = \mu(1-u^2)\left(\left(\frac{\mathrm{d}\tilde{n}}{\mathrm{d}h}\right)^2 + \frac{1}{(a+h)^2}\right), \tag{17}$$

$$\frac{\mathrm{d}\mu}{\mathrm{d}r} = 2\mu u\left(\frac{\mathrm{d}\tilde{n}}{\mathrm{d}h} + \frac{1}{a+h}\right) - \psi. \tag{18}$$





The integration of the adjoint variables requires knowing the spatial gradient of the modified refractive index at the query points $h$ along the ray trajectory. The refractive index profile was optimised using the two-stage optimisation procedure described by

Teh et al. (2022): forward tracing and backward tracing. The forward tracing stage involves integrating (10b) and (10c) using initial conditions $h_0$ and $u_0$ and evaluating the endpoint of the ray trajectory $h|_{r=r_f}$. The backward tracing stage then involves initialising the adjoint variables (13) and (14) at the termination position and integrating (17) and (18) in reverse along the same ray trajectory as in the forward tracing stage. The variation of the Lagrangian with respect to the refractive index is then

$$\delta\mathcal{L}|_{\tilde{n}} = \int_0^{r_f} \mu(1-u^2)\left(\frac{\mathrm{d}(\delta\tilde{n})}{\mathrm{d}h}\right)\mathrm{d}r, \tag{19}$$

which does not require an expensive Jacobian matrix solve to evaluate.

The quantity $\delta\tilde{n}$ is defined in terms of a linear interpolation scheme as

$$\tilde{n}(h,\boldsymbol{n}) = \boldsymbol{w}(h)\cdot\ln(\boldsymbol{n}) = \begin{bmatrix} w_0(h) & w_1(h) \end{bmatrix}\cdot\begin{bmatrix} \ln(n_0) \\ \ln(n_1) \end{bmatrix}, \tag{20}$$

where $n_0$ and $n_1$ are the refractive index values on the underlying grid directly below and above the sample point $h$ respectively. The weights $w_0(h)$ and $w_1(h)$ are inversely related to the vertical distance from the refractive index values $n_0$ and $n_1$

respectively. Therefore at each step in the integration of the adjoint variables the gradient of the Lagrangian with respect to the modified refractive index values on the underlying grid are calculated as

$$\frac{\mathrm{d}\mathcal{L}}{\mathrm{d}(\ln(n_0))} = \int_0^{r_f} \mu(1-u^2)\frac{\mathrm{d}w_0}{\mathrm{d}h}\mathrm{d}r, \tag{21a}$$

$$\frac{\mathrm{d}\mathcal{L}}{\mathrm{d}(\ln(n_1))} = \int_0^{r_f} \mu(1-u^2)\frac{\mathrm{d}w_1}{\mathrm{d}h}\mathrm{d}r, \tag{21b}$$

where the relation $\delta\tilde{n} = \boldsymbol{w}\cdot\mathrm{d}(\ln(\boldsymbol{n}))$ is employed. The geometry of the optimisation scheme at each step in the integration is

shown in Fig. 4.

The refractive index field was adjusted iteratively using a gradient descent scheme using a variation of the Adam optimiser (Kingma and Ba, 2014). The minimisation algorithm was terminated after a specified number of iteration steps. The gradient of the penalty function calculated using the adjoint state method was verified using finite differences over the elements of the refractive index profile, as shown in Fig. 5. The penalty function is highly sensitive to the surface refractive index value,

highlighting the importance of accurately knowing the surface conditions in order to constrain the optimised solution.

In summary, there are two principal observations required to retrieve atmospheric refractivity structure: the observed AoA of the radio broadcast (required to calculate the initial ray direction, $u_0$) and the position of the aircraft (required to determine $s_t$ and $h_t$). In the examples above we initialise the adjoint variable $\mu$ using the partial derivative of the penalty function with





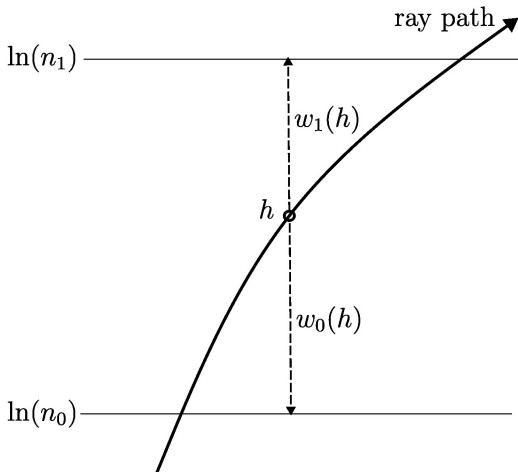

**Figure 4.** The geometric ray path intersecting the refractive index values $\ln(n_0)$ and $\ln(n_1)$ on the underlying grid.

respect to the ray direction $u$. However, in practice, we have no knowledge of the emission direction of the ADS-B signal.

Therefore we define the penalty function only in terms of the respective positions of the ray endpoint and the aircraft position. The adjoint term $\mu$ is therefore initialised at the termination position as $\mu|_{r=r_{\mathrm{f}}} = 0$.

## 2.4 Experimental arrangement

The observations were obtained using a prototype ADS-B interferometer installed on the Clee Hill weather radar tower in Shropshire, UK. The experimental setup is shown in Fig. 6. The vertical baseline of the interferometer was 13.86m, and each

of the two elements consisted of a stack of four Yagi-Uda antennas separated by 0.66 m. The stack reduced the beamwidth of each element, helping to mitigate the effects of signals reflected from nearby topography (multipath). The signals from the antenna stacks were connected to an Ettus X300 software-defined radio (SDR).

Approximately 263,500 refracted ADS-B transmissions were received between an AoA of $0.0°$ to $2.0°$ over an $\sim 80$-minute period between 08:36 and 09:58 UTC on the 22 September 2023. The synthetic and real retrievals were obtained using the

central $10°$ azimuthal sector of received transmissions. The distribution of the used transmissions is shown in Fig. 7a. The distribution in reported aircraft altitude within the chosen sector is shown in Fig. 7b. The north-east sector was chosen due to a suitable distribution in reported altitudes for the lower atmosphere to be adequately sampled. The nearby Watnall radiosonde station also allowed for a reference refractivity ground truth to be determined to compare retrieved refractivity profiles to.

The adjoint inversion algorithm was tested on synthetic data generated with ADS-B data recorded using the prototype

interferometer at Clee Hill, Shropshire. A subset of 5000 ADS-B transmissions were chosen randomly from within the central $10°$ azimuthal sector. To account for the ellipsoid shape of the Earth in the one-dimensional ray tracing model, the mean





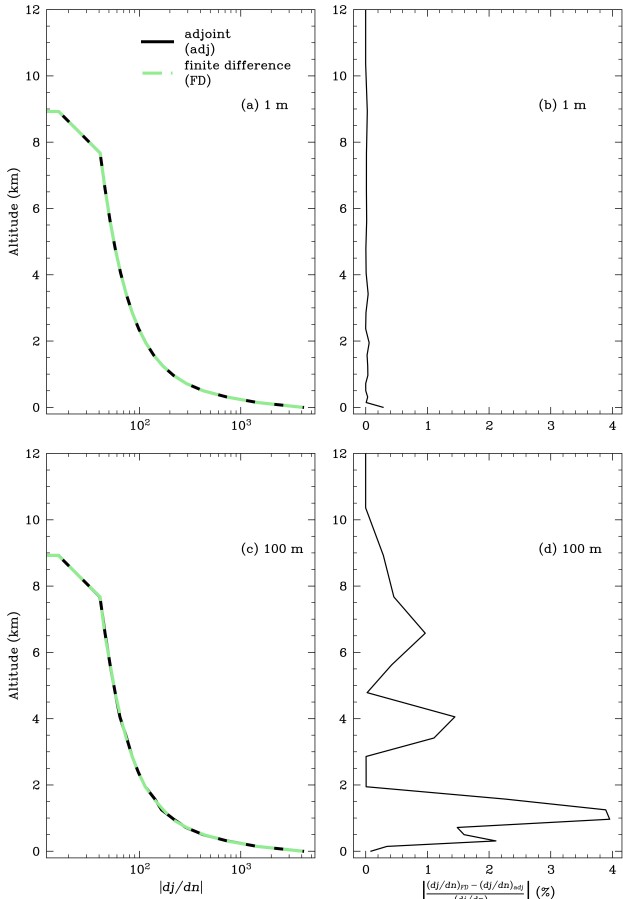

**Figure 5. Left panels:** The gradient of the penalty function with respect to the refractive index at each altitude point calculated using finite differences (green) and the adjoint state method (black) for (**a**) 1 m and (**c**) 100 m ray integration step size. The penalty function is highly sensitive to the refractive index values close to the surface. **Right panels:** The absolute relative difference between the gradients calculated using finite differences (FD) and the adjoint state method (adj) for (**b**) 1 m and (**d**) 100 m ray integration step size. The ray had an observed AoA of $0.11°$. The altitude and distance of the source aircraft were 11.2 km and 409.5 km respectively.

local radius of the ellipsoid at the location of the receiver is used as the effective radius of the Earth. The effective radius is approximately the radius of curvature of the ellipsoid at the location of the receiver, which ensures that contours of constant refractivity are approximately parallel to the curved surface of the ellipsoid. The height above the ellipsoid reported in the

ADS-B transmission was left unchanged. The coordinate transform is described in Appendix B. Radiosonde data from the nearby Watnall station in Nottinghamshire (position shown in Fig. 7a) was used to generate a refractivity profile to simulate the refraction of the subset of ADS-B transmissions. Three radiosonde soundings were used to generate the synthetic observations - 12:00 UTC 22 September 2023 (see Fig. 2), 12:00 UTC 18 July 2022 (during an intense heatwave) and 12:00 UTC 15 December 2022 (during a cold spell) (see Fig 8). The simulated samples of refracted ADS-B transmissions were injected





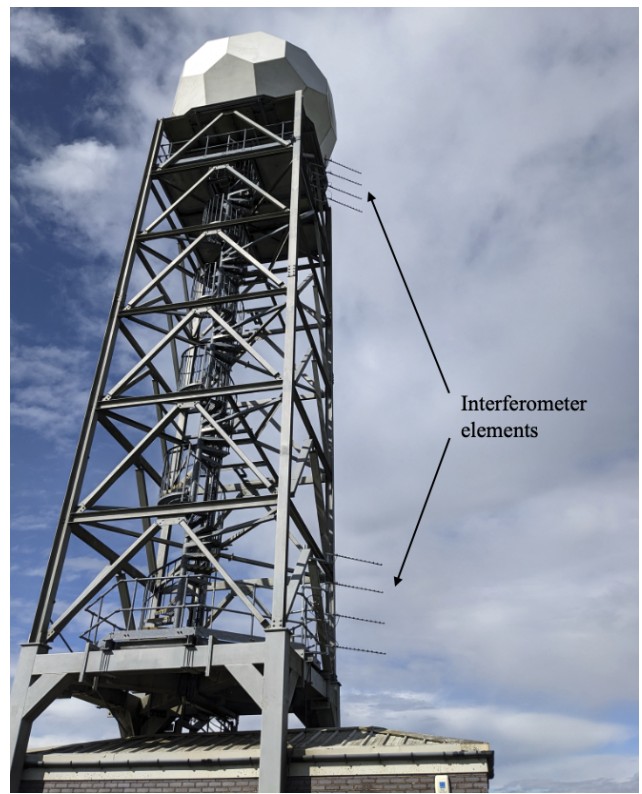

**Figure 6.** The interferometer mounted to the side of the Clee Hill weather radar tower.

with varying degrees of noise to model the impact of experimental uncertainties on the retrieved refractivity profiles. A prior (first-guess) refractivity profile was used to initial the penalty function, given by

$$N(h) = N_0 \exp\left(-\frac{h}{H}\right),\tag{22}$$

where $N_0$ is the refractivity at the position of the receiver (this was fixed to the radiosonde value during the optimisation procedure) and $H$ is the scale height in kilometres. Rays were traced backwards from the receiver position to the aircraft
using the known observed AoAs, as shown in Fig. 3. The total penalty function was then evaluated by summing together the penalty function for each ray. Since multipath contamination of the observed AoA measurement is thought to constitute the principal source of uncertainty in the retrieved refractivity profile, three observed AoA measurement noise test cases were investigated. The retrieval algorithm was run with a zero measurement noise case and two normally distributed noise test cases (with standard deviations of $0.01°$ and $0.05°$ respectively).





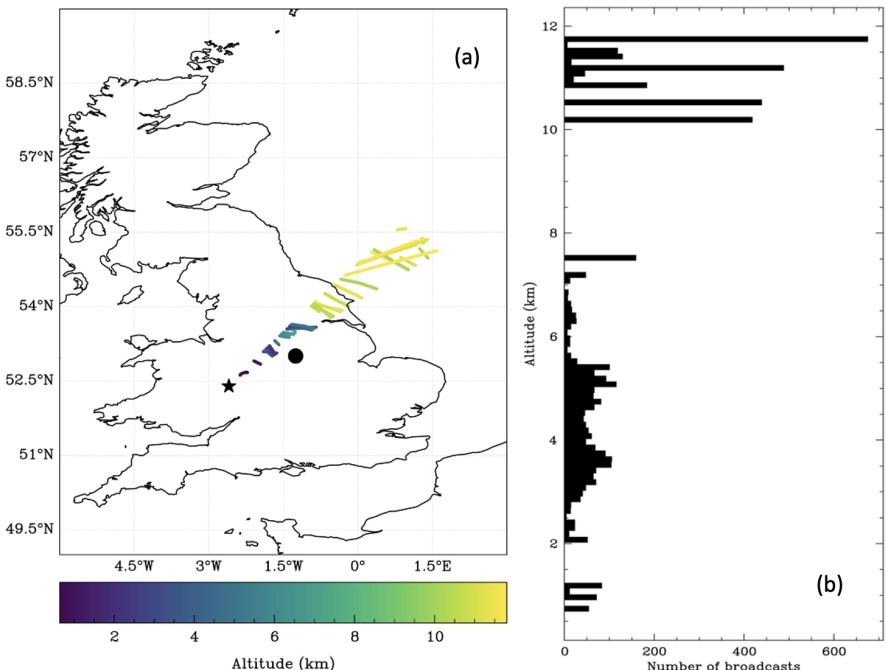

**Figure 7. (a)** The spatial distribution of the analysed aircraft transmissions colour-coded by reported altitude (km). The positions of the interferometer and the Watnall radiosonde station are indicated by the black star and black circle respectively. **(b)** Histogram of the distribution of broadcasts by reported altitude (km). The analysed aircraft transmissions were sampled from the central $10°$ azimuthal sector of received transmissions on the 22 September 2023.

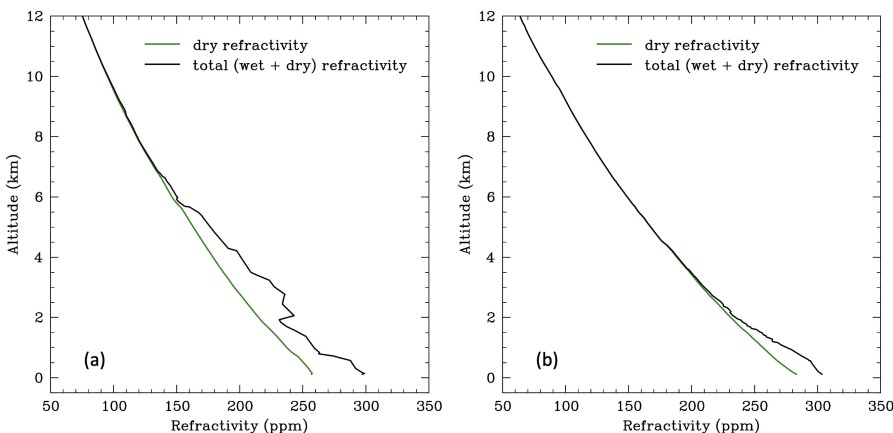

**Figure 8.** The total refractivity (black line) and dry refractivity (green line) versus altitude (in kilometres) retrieved using the 12:00 UTC 18 July 2022 **(a)** and 15 December 2022 **(b)** launch radiosonde data from Watnall respectively. The maximum altitude shown is 12 km.





**Table 1.** The RMS values for the initial refractivity profile and the retrieved refractivity profile (relative to the reference radiosonde profile) for different observed AoA noise standard deviations.

| Radiosonde data (12:00 UTC) | Initial RMS (ppm) | Retrieved RMS (ppm) | | |
|---|---|---|---|---|
| | | $0.00°$ | $0.01°$ | $0.05°$ |
| 22 September 2023 | 8.64 | 0.96 | 1.51 | 3.59 |
| 18 July 2022 | 9.77 | 1.81 | 3.44 | 8.39 |
| 15 December 2022 | 5.10 | 0.54 | 1.03 | 1.73 |

# 3 Results and analysis

## 3.1 Synthetic refractivity retrievals

The retrieved refractivity profile consisted of 30 levels on a logarithmically-spaced grid between an altitude of 0.575 km (height of receiver above the ellipsoid Earth model, estimated using a GNSS receiver on site) to an altitude of $\sim 13$ km. A ray step size of $\Delta r = 0.1$ km was used during the integration. A learning rate of $10^{-7}$ was used during the optimisation procedure. In order to better constrain the retrieved solution (particularly in the presence of noise) the minimum refractivity after every iteration during the optimisation procedure was fixed to be greater than or equal to the dry refractivity (i.e. assuming pressure and temperature are known and the relative humidity can never drop below 0%). The root mean squared (RMS) error in the synthetic retrieved refractivity profiles after convergence for the test cases using measurement noises of $0.00°$, $0.01°$ and $0.05°$ are shown in Table 1. The RMS in the retrieved profile increases as the standard deviation in the AoA noise increases. However, all retrieved profiles show improvement relative to the prior refractivity profile. A synthetic retrieval with no AoA measurement noise was also performed assuming the height of the receiver above the ellipsoid had an error of $\pm 5$ m and $\pm 10$ m. The synthetic retrieval assuming a receiver height error still showed significant improvement in the lower relative to the initial refractivity profile. However, there was little improvement above an altitude of $\sim 10$ km. The weakly constrained refractivity retrievals at high altitudes are explored in Section 4.

Figure 9a shows the synthetic retrieved vertical refractivity profile for the different observed AoA measurement noise cases using the 22 September radiosonde data. The initial refractivity profile was defined using (22) with a scale height $H = 8$ km. The percentage difference between the retrieved and the radiosonde-derived refractivity profile is shown in Fig. 9b.

The vertical refractivity profile for the 12:00 UTC 18 July 2022 radiosonde sounding is shown in Fig. 8a. The presence of a positive refractivity gradient at an altitude of approximately 2 km results in the largest deviation between the retrieved profile and the reference profile. In the presence of no measurement noise, the complex refractivity structure in this region can be resolved. However, in the presence of measurement noise exceeding a standard deviation of $0.01°$ the complex structure cannot be resolved. This is likely due to the inversion problem becoming ill-posed in the presence of observational noise. The synthetic retrievals using injected measurement noise with a standard deviation of $0.01°$ and $0.05°$ both did show improvements



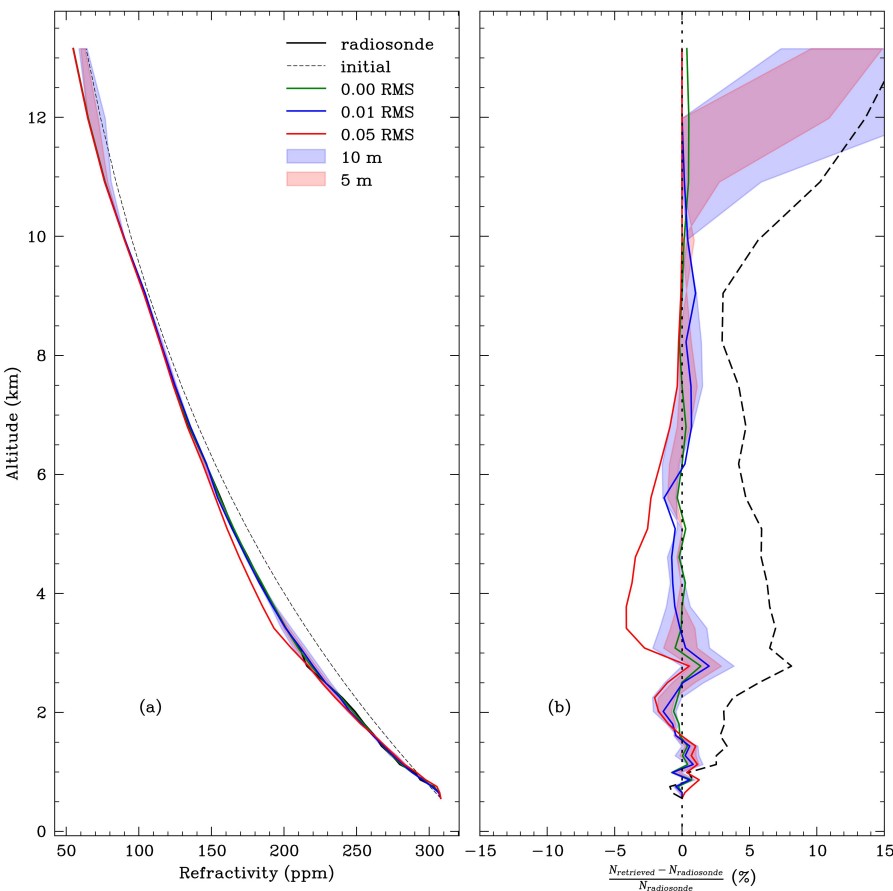

**Figure 9.** (a) The synthetic retrieved refractivity profile and the reference 22 September 2023 Watnall radiosonde profile for different synthetic error distributions injected into the observed AoA measurement. (b) The percentage difference between the synthetic retrieved refractivity profile and the reference radiosonde profile for different synthetic error distributions injected into the observed AoA measurement. The shaded regions indicate the uncertainty in the synthetic retrieved refractivity profile assuming no AoA measurement noise and a 5 m (red) and 10 m (blue) receiver altitude error respectively.

relative to the initial guess refractivity profile in the lower atmosphere, indicating that the noisy refraction observations still
include useful information. The vertical refractivity profile for the 12:00 UTC 15 December 2022 radiosonde sounding is shown in Fig. 8b. The refractivity profile shows little structure with only a very minor contribution from water vapour to the total refractivity, restricted to the first few kilometres of the surface. However, all three test noise cases show improvement in the synthetic retrieved refractivity structure relative to the initial guess profile.



## 3.2 Initial retrievals using observations

The retrieval algorithm was then tested on the observational data collected using the prototype ADS-B interferometer. The retrieved refractivity profiles were constrained by fixing the minimum refractivity to the dry refractivity. It is hoped that the technique will be able to resolve variations in humidity structure occurring on timescales of order an hour. Accordingly, the observed AoA measurements from one hour of the experiment were divided into four 15 minute periods and used to retrieve refractivity profiles. The observed AoA ranged between $0.0°$ and $2.0°$ and the horizontal angle range was chosen as the central

$10°$ of the sector shown in Fig. 7. Figure 10 shows the reported AoA as a function of horizontal angle for the four 15-minute periods of observational data. The black points (obs) indicate the reported AoA determined from the ADS-B positional data (i.e. the "true" reported flight paths). The blue points indicate the retrieved flight paths using the initial (prior) exponential refractivity profile. The green points indicate the retrieved flight paths using the optimised refractivity profile. The penalty function is minimised when the difference between the retrieved and true reported AoAs is minimised. The modelled reported

AoA (as shown by the green and blue points in Fig. 10) provides a direct comparison with observable quantities (the reported AoA determined using the received ADS-B transmission). The initial and retrieved modelled reported AoAs show ripple-like structure due to significant multipath. To model the reported AoA using a given refractivity profile, the ray is traced using the measured AoA (as shown in Fig. 3). Since the measured AoA is noisy, the modelled reported AoA will share the similar noisy structure. The impact of multipath on the retrieved refractivity profiles is explored below. However, despite the presence of

noise in the observed AoA measurement, the retrieved reported AoA generally converges towards the true reported AoA. The corresponding refractivity profiles for each 15-minute sample of observations are shown in Fig. 11. The reference refractivity profile was determined using the 12:00 UTC Watnall radiosonde launch on the 22 September 2023. The refractivity value at the altitude of the receiver was fixed to the radiosonde value at an altitude of 0.575 km. The refractivity profiles retrieved using observational data shows agreement with reference profile radiosonde profile in the lower atmosphere up to an altitude of $\sim 5$

km. At higher altitudes, the refractivity profiles show significant variability compared to lower altitudes, likely due to a lack of constraints here (a change in refractivity at higher altitudes has less impact than a change in refractivity at lower altitudes). The potential systematic uncertainties in the observations suggested by these results are explored in Section 4.

        The difference between the true and initial modelled reported AoA and the difference between the true and retrieved modelled reported AoA as a function of surface distance is shown in Fig. 12. The distribution in the difference between the true

and initial modelled reported AoA and the difference between the true and retrieved modelled reported AoA is shown in Fig. 13. The mean of the reported AoA difference decreased from $\sim 0.034°$ to $\sim 0.009°$ and and the standard deviation increased from $\sim 0.023°$ to $\sim 0.024°$ before and after the retrieval respectively. The standard deviation in the reported AoA exceeds the target accuracy of $0.01°$, meaning that complex refractivity structure is unlikely to be resolved. The modelled reported AoA using the initial refractivity profile is larger than the true reported AoA, indicating the initial atmosphere is less refractive than

the true atmosphere. The retrieved atmosphere results in apparent weaker refraction of ADS-B signals from nearby aircraft and apparent stronger refraction for signals originating from more distant aircraft. The apparent variability in the observed refraction of ADS-B signals originating from aircraft at varying distances is likely to be partially a consequence of observational





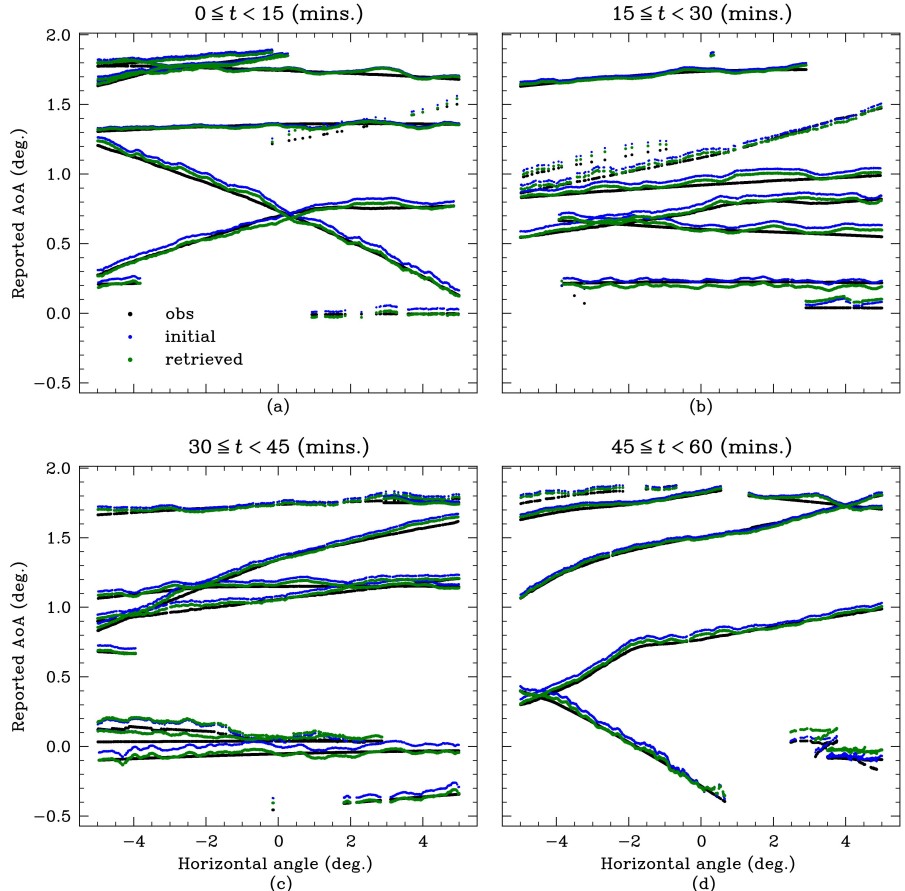

**Figure 10.** The reported AoA as a function of horizontal angle. The black points indicate the reported AoA derived from the decoded ADS-B transmissions. The blue points indicate the modelled reported AoA determined from tracing rays through the initial (first-guess) refractivity profile. The green points indicate the modelled reported AoA determined from tracing rays through the retrieved (optimised) refractivity profile.

noise. The noise is visible in Fig. 10 as a ripple-like structure in the flight paths. This noise is correlated with the foreground environment and is not truly random, which likely results in a bias in the retrieved refractivity profiles. Potential sources of uncertainty, including multipath interference and instrumental errors are described in Section 4.

## 4 Discussion

Initial tests with synthetic data and the case study using actual measurements have revealed possible weaknesses in the retrieval method. Sources of experimental uncertainty that require further attention are discussed in the subsections below.



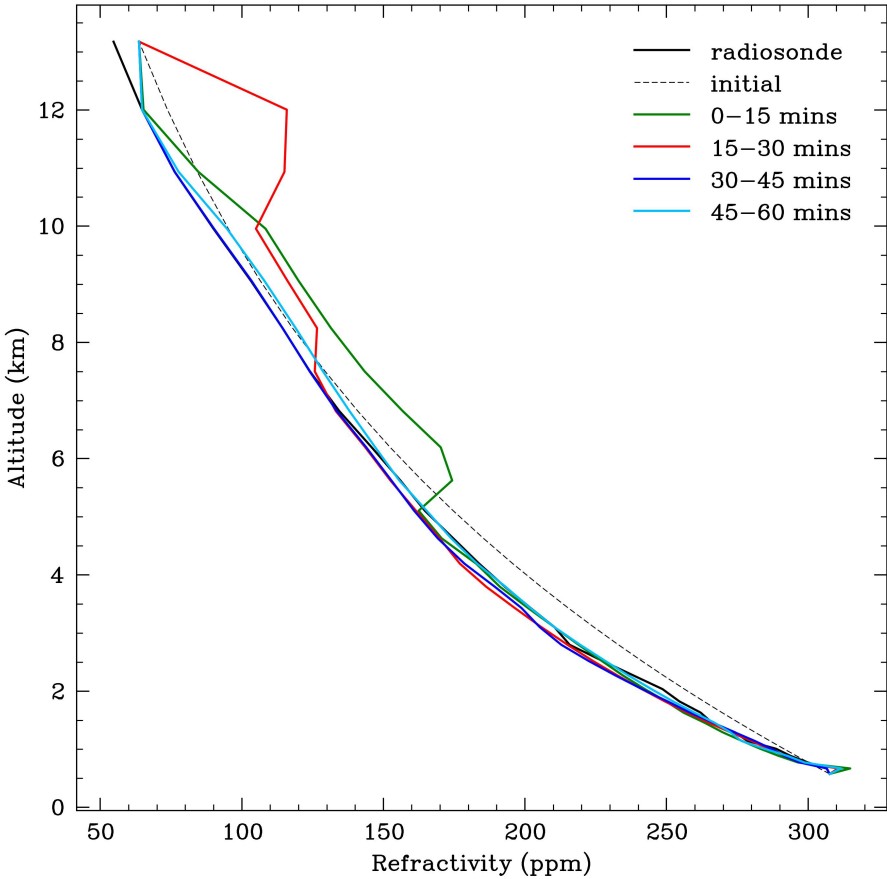

**Figure 11.** The 12:00 UTC 22 September 2023 Watnall radiosonde refractivity profile (solid black), retrieved (solid colour) and initial (dashed) refractivity profiles.

## 4.1 Assumption of a horizontally homogeneous refractivity field

The SODE ray tracing model was derived with the assumption of spherical symmetry (the refractivity varies only with radial distance and is horizontally homogeneous). Therefore only one-dimensional refractivity profiles can be derived using the adjoint model presented here. For rays with negative elevation angles, this assumption will likely break down rapidly as the ray propagates through the highly variable planetary boundary layer. In this study only ADS-B transmissions with an observed AoA $\geq 0°$ were used. As the rays generally increase in altitude with distance due to the curvature of the Earth, the assumption of

spherical symmetry is therefore assumed to be reasonable. However, significant horizontal variability in refractivity can occur along frontal systems, which can invalidate the spherical symmetry assumption. An occluded front was present to the south east of the UK during the observation period. However, no significant frontal activity was present within the observation sector. The weather during the observation period was partially cloudy, with patchy low cloud present around Clee Hill. Local variations in





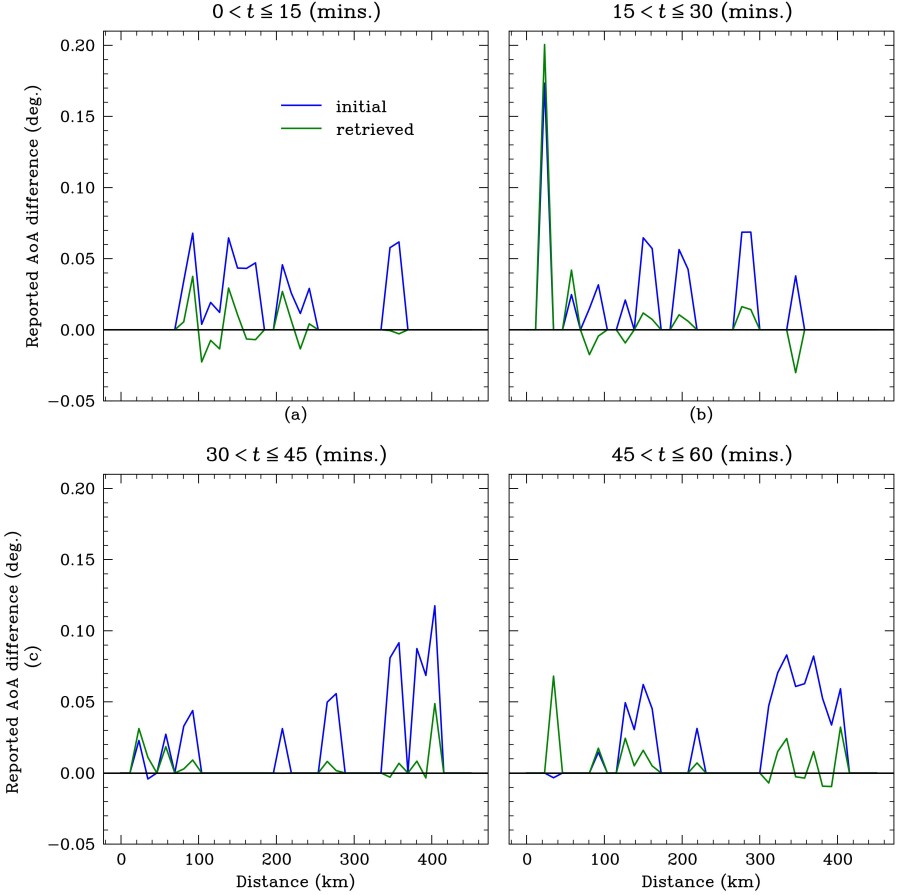

**Figure 12.** The difference between the true (determined from the ADS-B transmissions) and initial modelled reported AoA (blue line) and the difference between the true and retrieved modelled reported AoA (green line) as a function of distance.

the refractivity near the receiver could contribute to uncertainty in the refractivity retrievals. However, the significant multipath

contamination of the observed AoA measurement was expected to be the dominant source of uncertainty in the retrieved refractivity profiles.

To assimilate AoA measurements influenced by significant horizontal refractivity gradients, a two-dimensional ray tracing model and its adjoint will be required. It is uncertain how well constrained a two-dimensional refractivity retrieval will be using a single ADS-B interferometer. A network of ADS-B interferometers across the UK could potentially allow for two- or

three-dimensional refractivity profiles to be obtained.




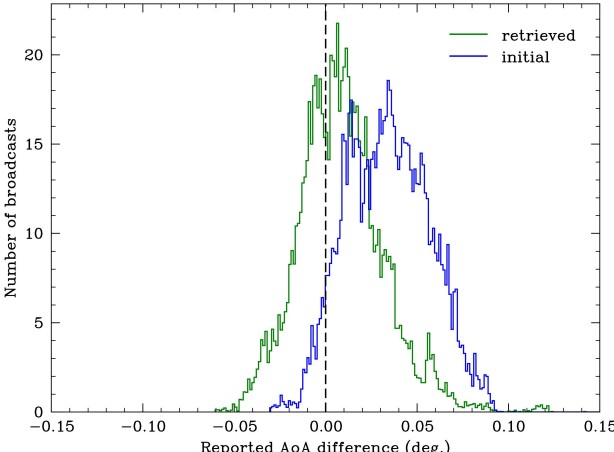

**Figure 13.** The distribution in the reported AoA difference for all observations using the initial (blue) and retrieved (green) refractivity profiles. The mean difference in the reported AoA decreased from $\sim 0.034°$ to $\sim 0.009°$ for the initial and retrieved refractivity profiles respectively. The standard deviation in the reported AoA difference increased from $\sim 0.023°$ to $\sim 0.024°$ for the initial and retrieved refractivity profiles respectively.

## 4.2 Sensitivity of the retrievals to surface refractivity

The retrieved refractivity profile is extremely sensitive to the surface refractivity (as shown in Fig. 5). Uncertainty in the measured refractivity here will result in erroneous retrievals, therefore an accurate in-situ measurement of refractivity will likely be required to constrain the retrieved solution. Accurate on-site refractivity measurements could be obtained using a permanent weather station. The bending of radio signals is dominated by the refractivity gradient, rather than the absolute refractivity (since the refractive index of air is approximately unity even near the surface). Any uncertainty in the surface refractivity introduces a constant offset throughout the entire retrieved profile, however the retrieved refractivity gradient remains unchanged. The surface refractivity uncertainty could be addressed by instead assimilating refractivity gradients into NWP models, which would reduce the need for accurate surface refractivity measurements.

## 4.3 Multipath

Estimating the observed AoA to an uncertainty of $0.01°$ requires accurate measurements of the phase difference across the interferometer elements. A significant source of uncertainty in this measurement is due to the presence of multipath. At grazing elevation (observed AoA $\lesssim 2.0°$) the direct and reflected ADS-B signal are received in rapid succession, resulting in mutual interference. The multipath results in ripple-like structures in the observed flight paths of the aircraft, as shown in Fig. 10. The uncertainties in the observed AoA measurements are highly correlated and dependent on the local terrain, which makes estimating the observation error covariance matrix very difficult. Efforts to minimise the impacts of multipath have included beam forming by stacking multiple antennas on each element of the interferometer (see Fig. 6) and increasing the sampling



rate. The beam forming technique aims to deepen and control the position of the antenna nulls to minimise the sensitivity to foreground reflections. Increasing the sampling rate allows for phase measurements to be extracted from the direct signal before

the arrival of the reflected signal. The sampling rate used for the experiment was 10 MHz, which was able to strongly suppress multipath originating from foreground reflections with a delay longer than $\sim 100$ ns (excess path length $\sim 30$ m). Multipath due to foreground reflections with a path excess shorter than this can only be partially suppressed using the current ADS-B interferometer. Future experiments will aim to further suppress the multipath by increasing sampling rate and modifying the beam pattern of the stacks. Both techniques have resulted in significant improvements in the accuracy of the observed AoA

measurement.

### 4.4   Instrumental uncertainties

Instrumental errors may include the assumption that a measured phase difference of zero corresponds to an observed AoA of $0.00°$. A slight offset between the upper and lower elements of the interferometer could introduce uncertainty in this assumption. The reported position of the aircraft is also limited by the reporting resolution (on the order $\sim 10$ m, Stone and Kitchen

(2015)), however the impact of positional uncertainty on the observed AoA was expected to be minor due to the large distances to aircraft (Lewis et al., 2023a). Uncertainty in the altitude of the receiver will introduce a systematic error in the retrieved refractivity profiles. Figure 9 shows the impact of uncertainty in the height of the receiver on the retrieved refractivity profiles. The height uncertainty results in significant variability in refractivity at high altitudes. The weakly constrained refractivity retrievals above an altitude of $\sim 5$ km for some of the retrieved profiles in Fig. 11 suggest significant observational noise. This

suggests a potential systematic error in the observational data. However, since the high altitude variability was only present in some of the retrieved profiles, it may suggest that the source of the uncertainty may not be entirely systematic. The gradient of the penalty function with respect to refractivity at higher altitudes is significantly lower than at lower altitudes, as shown in Fig. 5. Introducing additional constraints to the penalty function to penalise large departures from the reference refractivity profile may reduce the presence of unphysical refractivity structures. In practice, a one-dimensional variational (1D-Var) data

assimilation scheme would prevent unphysical retrievals. However, a good understanding of the observational uncertainties would be required.

     The mean altitude of the midpoint between the elements of the interferometer above the ellipsoid Earth model was estimated to be 575 m using a GNSS receiver located on site. The standard deviation of the measured altitude was estimated to be 2 m. Multipath can result in correlated uncertainties in the observed AoA measurements, potentially resulting in biased retrievals.

## 5   Conclusions

A new method has been developed to retrieve refractivity profiles from measured AoA of radio signals from aircraft. The method incorporates the SODE ray tracing model (Zeng et al., 2014) and was tested on both synthesised AoA data (to explore sensitivity to measurement noise) and real AoA data measured with an experimental interferometer. Three AoA measurement noise cases were simulated, consisting of no noise and two normally distributed noise cases (with standard deviations of $0.01°$



and $0.05°$ respectively) injected into the observed AoA measurements. The impact of each of the injected observed AoA measurement uncertainty cases on the retrieved vertical refractivity profiles was investigated. The convergence of the penalty function was impacted negatively and the RMS in the retrieved refractivity profile increased as the standard deviation in the injected (normally distributed) noise increased. The synthetic retrievals using a AoA noise standard deviation of $0.01°$ where of comparable accuracy to vertical refractivity profile retrievals using GNSS radio occultation sounding (Anthes et al., 2022), airborne GNSS radio occultation sounding (Murphy et al., 2015) and GNSS troposphere tomography (Wu et al., 2014; Trzcina and Rohm, 2019). When the algorithm was tested on real measurements in four separate 15-minute periods, the retrieved profiles showed significant variability at high altitudes, but good convergence towards the reference radiosonde profile in the lowest few kilometres above the surface. Sources of uncertainty were explored and future work was suggested, such as improvements in the instrument design to minimise multipath contamination and the development of a two-dimensional adjoint model to capture horizontal variations in refractivity.

**Appendix A**

A more general constrained optimisation problem can be expressed in terms of the state variables $\boldsymbol{v}$ and the control variables $\boldsymbol{p}$ as

$$\min_{\boldsymbol{p}} \quad f(\boldsymbol{v}(\boldsymbol{p})) \tag{A1a}$$

$$\text{subject to} \quad \boldsymbol{g}(\boldsymbol{v},\boldsymbol{p}) = 0, \tag{A1b}$$

where $f$ is the scalar functional we seek to minimise and $\boldsymbol{g}$ are the constraints on the optimisation (equivalent to the constraint equations (10b)-(10e) in the original optimisation problem) respectively. In a data assimilation approach, $\boldsymbol{g}$ is sometimes referred to as the forward model and describes the physics of the system. We can convert the constrained optimisation problem described by (A1) into an unconstrained optimisation problem by introducing two vectors of slack variables known as Lagrange multipliers. The Lagrangian, $\mathcal{L}$, can be defined in terms of $f$ and the constraints $\boldsymbol{g}$ as

$$\mathcal{L}(\boldsymbol{v},\boldsymbol{p},\boldsymbol{\Lambda}) \equiv f(\boldsymbol{v}) - <\boldsymbol{\Lambda}^T, \boldsymbol{g}(\boldsymbol{v},\boldsymbol{p})>, \tag{A2}$$

where $\boldsymbol{\Lambda}$ is a vector of adjoint variables associated with the constraints $\boldsymbol{g}$ and $<\cdot,\cdot>$ is the inner product. The minimum of the Lagrangian, $\mathcal{L}$, and the scalar functional, $f$, will be equivalent when all the variables except for $\boldsymbol{p}$ are stationary points (that is, variations of the Lagrangian with respect to $\boldsymbol{v}$ and $\boldsymbol{\Lambda}$ are equal to zero). The total derivative of $\mathcal{L}$ with respect to $\boldsymbol{p}$ is written as

$$\frac{d\mathcal{L}}{d\boldsymbol{p}} = \frac{\partial f}{\partial \boldsymbol{v}}\frac{d\boldsymbol{v}}{d\boldsymbol{p}} - \frac{d\boldsymbol{\Lambda}^T}{d\boldsymbol{p}}\boldsymbol{g} - \boldsymbol{\Lambda}^T\left(\frac{\partial \boldsymbol{g}}{\partial \boldsymbol{v}}\frac{d\boldsymbol{v}}{d\boldsymbol{p}} + \frac{\partial \boldsymbol{g}}{\partial \boldsymbol{p}}\right). \tag{A3}$$





Since the constraints $\boldsymbol{g} = 0$, the second term on the right hand side of (A3) can be dropped. The expression then becomes

$$\frac{\mathrm{d}\mathcal{L}}{\mathrm{d}\boldsymbol{p}} = \frac{\mathrm{d}\boldsymbol{v}}{\mathrm{d}\boldsymbol{p}}\left(\frac{\partial f}{\partial \boldsymbol{v}} - \boldsymbol{\Lambda}^T \frac{\partial \boldsymbol{g}}{\partial \boldsymbol{v}}\right) - \boldsymbol{\Lambda}^T \frac{\partial \boldsymbol{g}}{\partial \boldsymbol{p}}. \tag{A4}$$

In a data assimilation scheme, the Jacobian matrix $\mathrm{d}\boldsymbol{v}/\mathrm{d}\boldsymbol{p}$ describes the sensitivity of the trajectory of the state variables $\boldsymbol{u}$ in

the forward model to changes in the control variables $\boldsymbol{p}$. For large numbers of control variables the cost of computing $\mathrm{d}\boldsymbol{v}/\mathrm{d}\boldsymbol{p}$ quickly becomes very high. However, since $\boldsymbol{g}(\boldsymbol{v},\boldsymbol{p}) = 0$ in our original problem statement (A1), we can freely choose the Lagrange multiplier $\boldsymbol{\Lambda}$. If we choose $\boldsymbol{\Lambda}$ such that

$$\boldsymbol{\Lambda}\left(\frac{\partial \boldsymbol{g}}{\partial \boldsymbol{v}}\right)^T = \left(\frac{\partial f}{\partial \boldsymbol{v}}\right)^T, \tag{A5}$$

the first term on the right hand side of (A4) is zero and the need to calculate the Jacobian matrix $\mathrm{d}\boldsymbol{v}/\mathrm{d}\boldsymbol{p}$ is avoided (Zou et al.,

1997; Gill et al., 1981).

The penalty function for a single ray, $j$, can be augmented with the known dynamics of the system using Lagrange multipliers:

$$\mathcal{L} = j - \int_0^{r_\mathrm{f}} \psi(\dot{h} - u)\mathrm{d}r - \int_0^{r_\mathrm{f}} \mu\left(\dot{u} - (1 - u^2)\left(\frac{1}{n}\frac{\mathrm{d}n}{\mathrm{d}h} + \frac{1}{a+h}\right)\right)\mathrm{d}r, \tag{A6}$$

where $\dot{\psi}$ and $\dot{\mu}$ are the derivatives of the adjoint variables $\psi$ and $\mu$ with respect to $r$ respectively. The stationary points of the

Lagrangian with respect to $h$ and $u$ are found as

$$\delta\mathcal{L}|_h = \delta j|_h - \int_0^{r_\mathrm{f}} \psi\delta\dot{h}\mathrm{d}r - \int_0^{r_\mathrm{f}} \mu(1 - u^2)\left(\left(\frac{1}{n}\frac{\mathrm{d}n}{\mathrm{d}h}\right)^2 - \frac{1}{n}\frac{\mathrm{d}^2 n}{\mathrm{d}h^2} + \frac{1}{(a+h)^2}\right)\delta h\mathrm{d}r = 0, \tag{A7}$$

$$\delta\mathcal{L}|_u = \delta j|_u + \int_0^{r_\mathrm{f}} \psi\delta u\mathrm{d}r - \int_0^{r_\mathrm{f}} \mu\delta\dot{u}\mathrm{d}r - \int_0^{r_\mathrm{f}} 2\mu u\left(\frac{1}{n}\frac{\mathrm{d}n}{\mathrm{d}h} + \frac{1}{a+h}\right)\delta u\mathrm{d}r = 0. \tag{A8}$$
.

Integrating (A7) and (A8) by parts

$$\delta\mathcal{L}|_h = \delta j|_h - [\psi\delta h]_0^{r_\mathrm{f}} + \int_0^{r_\mathrm{f}} \dot{\psi}\delta h\mathrm{d}r - \int_0^{r_\mathrm{f}} \mu(1 - u^2)\left(\left(\frac{1}{n}\frac{\mathrm{d}n}{\mathrm{d}h}\right)^2 - \frac{1}{n}\frac{\mathrm{d}^2 n}{\mathrm{d}h^2} + \frac{1}{(a+h)^2}\right)\delta h\mathrm{d}r = 0, \tag{A9}$$

$$\delta\mathcal{L}|_u = \delta j|_u + \int_0^{r_\mathrm{f}} \psi\delta u\mathrm{d}r - [\mu\delta u]_0^{r_\mathrm{f}} + \int_0^{r_\mathrm{f}} \dot{\mu}\delta u\mathrm{d}r - \int_0^{r_\mathrm{f}} 2\mu u\left(\frac{1}{n}\frac{\mathrm{d}n}{\mathrm{d}h} + \frac{1}{a+h}\right)\delta u\mathrm{d}r = 0. \tag{A10}$$





The adjoint variables at $r = 0$ are $\psi_{r=0} = 0$ and $\mu_{r=0} = 0$. Therefore

$$\psi_{r=r_{\mathrm{f}}} = \frac{\partial j}{\partial h}, \tag{A11}$$

$$\mu_{r=r_{\mathrm{f}}} = \frac{\partial j}{\partial u}, \tag{A12}$$

$$\frac{\mathrm{d}\psi}{\mathrm{d}r} = \mu(1 - u^2)\left(\left(\frac{1}{n}\frac{\mathrm{d}n}{\mathrm{d}h}\right)^2 - \frac{1}{n}\frac{\mathrm{d}^2 n}{\mathrm{d}h^2} + \frac{1}{(a+h)^2}\right), \tag{A13}$$

$$\frac{\mathrm{d}\mu}{\mathrm{d}r} = 2\mu u\left(\frac{1}{n}\frac{\mathrm{d}n}{\mathrm{d}h} + \frac{1}{a+h}\right) - \psi. \tag{A14}$$

**Appendix B**

The reported AoA of the aircraft at the location of the observer is computed in terms of the geodetic Cartesian coordinates of the observer. The geodetic Cartesian coordinates of the observer $(X_o, Y_o, Z_o)$ can be calculated using the geodetic coordinates
latitude ($\phi_o$), longitude ($\lambda_o$) and height ($h_o$) of the observer as

$$X_o = (N(\phi_o) + h_o)\cos(\phi_o)\cos(\lambda_o), \tag{B1}$$

$$Y_o = (N(\phi_o) + h_o)\cos(\phi_o)\sin(\lambda_o), \tag{B2}$$

$$Z_o = (N(\phi_o) + h_o)\sin(\phi_o), \tag{B3}$$

where $N$ is the prime vertical radius, defined in terms of the geodetic latitude $\phi$ as

$$N(\phi) = \frac{a}{\sqrt{1 - e^2\sin^2(\phi)}}, \tag{B4}$$

and $e^2 = 1 - a^2/b^2$, where $a = 6378.137$ km and $b = 6356.75231425$ km respectively (Decker, 1986; Hofmann-Wellenhof et al., 1997). The ellipsoid Earth geometry is shown in Figure B1.



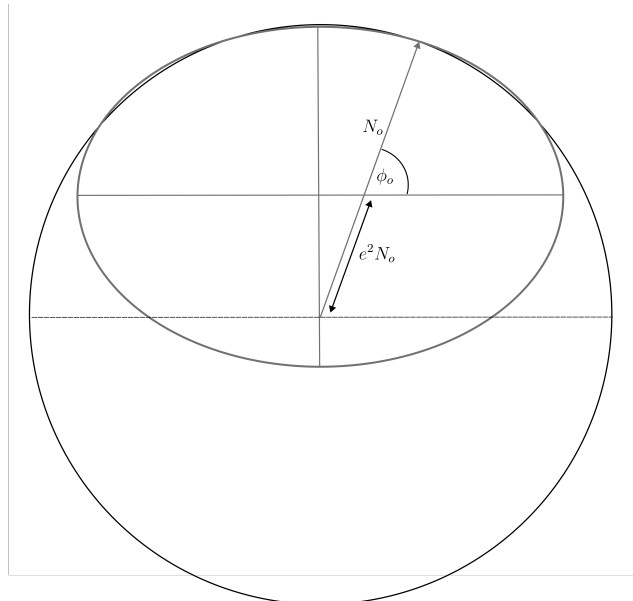

**Figure B1.** The ellipsoid Earth model (thick grey line). The sphere with the radius equal to the prime vertical radius is shown.

The geocentric coordinates of the aircraft $(X'_p, Y'_p, Z'_p)$ are given by

$$X'_p = (N(\phi_p) + h_p)\cos(\phi_p)\cos(\lambda_p), \tag{B5}$$

$$Y'_p = (N(\phi_p) + h_p)\cos(\phi_p)\sin(\lambda_p), \tag{B6}$$

$$Z'_p = (N(\phi_p)(1 - e^2) + h_p)\sin(\phi_o), \tag{B7}$$

where $\phi_p$, $\lambda_p$ and $h_p$ are the geodetic latitude, geodetic longitude and altitude of the aircraft above the ellipsoid respectively. The coordinates of the aircraft in terms of the geodetic coordinates of the observer are

$$X_p = X'_p, \tag{B8}$$

$$Y_p = Y'_p, \tag{B9}$$



$$Z_p = Z_p' + e^2 N(\phi_o)\sin(\phi_o). \tag{B10}$$

The geometry is shown in Figure B2. Defining the position vectors of the observer and aircraft as $\boldsymbol{r}_o = (X_o, Y_o, Z_o)$ and $\boldsymbol{r}_p = (X_p, Y_p, Z_p)$ respectively, the vector between the position of the observer and the aircraft is

$$\Delta\boldsymbol{r} = \boldsymbol{r}_p - \boldsymbol{r}_o. \tag{B11}$$

Therefore the zenith angle, $\Omega$, of the aircraft as seen by the observer can be written in terms of the unit vectors

$$\Omega = \cos^{-1}(\Delta\hat{\boldsymbol{r}} \cdot \hat{\boldsymbol{r}}_o). \tag{B12}$$

The reported AoA of the aircraft is therefore equal to $\pi/2 - \Omega$. The mean local radius of curvature of an ellipsoid at a given point and azimuth direction is given by (Jekeli, 2006) as

$$R_{\text{roc}} = \left( \frac{\sin^2(\alpha)}{N} + \frac{\cos^2(\alpha)}{M} \right)^{-1}, \tag{B13}$$

where $M$ is the radius of curvature of the meridian ellipse, given by

$$M = \frac{a(1-e^2)}{(1 - e^2\sin^2(\phi_o))^{\frac{3}{2}}}. \tag{B14}$$

The radius of spherical Earth used in the ray tracing model was assumed to be the local radius of curvature of the ellipsoid at the position of the receiver, $R_{\text{roc}} \approx 6383.57$ km (assuming an azimuth $\alpha = 45°$ north east).

*Code and data availability.* The ray tracing and adjoint inversion code are provided in the Github repository available at https://github.com/osl202/ADS_B_refractivity. The ADS-B data used in this study are available at https://doi.org/10.5281/zenodo.12783749. The radiosonde data was provided the University of Washington (see http://weather.uwyo.edu/upperair/sounding.html).

*Author contributions.* Conceptualisation and instrument design, CB and MK; adjoint model design, OL; software, OL and CB; formal analysis, OL and CB; investigation, OL; data curation, CB; writing—original draft preparation, OL; writing—review and editing, MK, CB, 470 NEB, EKS; visualisation, OL; supervision, CB, MK, NEB, EKS. All authors have read and agreed to the published version of the manuscript.

*Competing interests.* The contact author has declared that none of the authors have any competing interests.





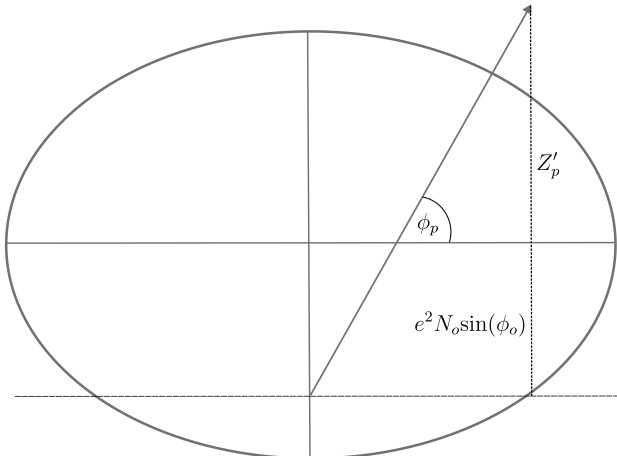

**Figure B2.** The $Z$-coordinate of the aircraft in terms of the geodetic coordinates of the observer.

*Financial support.* This work was supported by the Met Office, the University of Exeter and the Harry Otten Foundation

*Acknowledgements.* The authors would like to thank Met Office engineers Mike Protts and Paul Eadie for their support with the installation
of the interferometer on the Clee Hill weather radar tower. The authors would also like to thank Matthew Browning (University of Exeter) for
the very helpful discussion on the adjoint method. This project has made use of the following Python packages: PANDAS (Wes McKinney,
2010), NUMPY (Harris et al., 2020), MATPLOTLIB (Hunter, 2007), SMPLOTLIB (Li, 2023), CARTOPY (Met Office, 2010 - 2015)
and IRIS (Iris contributors).





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
