# Peer review of "Retrieving tropospheric refractivity structure using interferometry of aircraft radio transmissions"

_EGUsphere, 2024_

## Author Response (AR1)

**Response to Referee 1:**

- Line 73, about the limitation of the technique to measure AoA above 2 degrees: Is this limitation due to exceeding 1 cycle (1 wavelength) of delay between the top and the bottom elements of the interferometer? Or/and due to the way it is produced (angle of conjugate product of both signals)? If it is due to ambiguity due to a number of integer wavelengths in the relative delay, this could be overcame by resolving this number of integer cycles, which maybe could be achieved with code delays (if the bandwidth of the ABS-D is broad enough – I do not know), and/or by cross-correlating both signals searching for the maximum, instead of measuring only the phase of the complex conjugate. Furthermore, if the ABS-D system uses more than one frequency, combining information from both could also help resolving this ambiguity.

The limitation to the measured AoA being below an elevation of 2 degrees is due to significant bending of the signal occurring only at grazing incidence.  Signals can be received at higher elevations, although the influence of refraction is significantly smaller and less variable.  We use the expected AoA of the broadcasts (using a radiosonde refractivity profile nearby in time) to solve the AoA ambiguity arising from wrapped phases (i.e. to figure out the correct multiple of 2pi to add to the measured phase). Since the measured AoA typically only deviates from the predicted AoA by a few/several hundredths of a degree, this is a very reliable method. A disagreement greater than 0.57 deg. would be required to cause problems. Even in this case, the continuity of the flightpaths could be used to infer the correct solution.

- Did the authors analyse areas in Europe where the 2 deg topographic constraint would apply? Are there many zones where the interferometers could be located? Or very few?

No specific regions in Europe were analysed except for the Clee Hill site in Shropshire, UK.  Ideally, the interferometer would be located with a clear view of the horizon with airports visible (to allow ADS-B transmissions to be received throughout the extent of the lower atmosphere).  A single receiver on a hill-top site can receive ADS-B signals from over 500 km away, so the requirement for a dense network is reduced.

- More about measurement of AoA: are the Dopplers induced by the aircraft motion affecting the AoA? Are these Doppler shifts accounted for somewhere in the processing chain?

The doppler shift expected for aeroplane speeds is too small to have an effect of any importance on the received frequency (and therefore on the measured AoA).

- Line 79, '… beta is the incident AoA …' : as mentioned for epsilon later on, this angle tends to be called elevation angle rather than incidence (incidence would be the complementary to beta).

The AoA has been defined as the elevation angle of the signal for consistency.

- Line 100: the techniques implemented here (simulation and retrieval) are all based on geometric optics. Have the authors checked potential alternatives using wave optics?

No specific analysis using wave optics have been performed in this study, although work has been started to model the impact of atmospheric multipath and ducting and what signatures this might have in the observational data (the principal impact being the violation of the proportional relationship between the phase difference and the AoA, as multiple rays arrive at the receiver from the same source).

- Line 119, equation in-line $\epsilon_i = \sin^{-1} (u_i)$: it would help having this equation in a separate line and with a number. This would help understanding equation 10b and text in line 134.

This has been corrected.

- Figure 3: it would help a lot to have indication in the graphic about h_i, \epsilon_i and s_i (as used in equation mentioned above). This is specially true for the variable 's', as the explanation in lines 116-117 is not clear.

Figure 3 has been updated for clarity.

- Line 135: 'The initial position and direction… by h_0 and beta respectively': the equations above do not use beta but u_0, perhaps add in brackets the link between beta and u_0.

An added statement that u_0 = sin(beta) has been added.  When integrating the ray path, beta is the initial (observed) AoA of the signal, and throughout the ray path the elevation angle above the local horizon is epsilon.

- Figure 5: the plots on the right-column of this figure are not discussed. The sensitivity to bottom layers is deduced from the left-column. What is the right-column informing about? About the along-ray resolution needed to be applied? (it seems to require better than 100 m, but later it is said that the actual processing is done at 100 m)

The right-hand side demonstrates the convergence of the value of the gradient computed using finite differences and the adjoint state method.  A trade-off between the accuracy of the gradient calculation and the computational cost of integrating thousands of rays was needed.  The following has been added to the discussion:

"The right panels of Fig. 5 show the magnitude of the relative difference between the gradients calculated using finite differences and the adjoint state method for 1 m and 100 m ray integration step sizes respectively.  The value of the gradients computed using finite differences and the adjoint state method converge as the ray step size decreases, however, the computational cost of integrating rays with a very small step size increases significantly.  For this study the 100 m integration step size was used as the gradient calculation was sufficiently accurate without being prohibitively expensive to compute for thousands of rays."

Figure 7a: this shows only the +-5 deg azimuth around the antenna boresight. The configuration of the antenna arrays shown in the picture does not seem to limit much on azimuth (but elevation). Therefore, it is likely that the interferometer was able to collect data from a broader range of directions. I think it would be interesting to show them all, too, as a way to illustrate the potential of the technique from a single interferometer. For example, add a panel to the left of panel (a), showing all the measurements, then the central would be the current (a) with only those selected for this study.

A panel has been added to the left of the original Figure 7(a) showing the entire observed region.

Caption of Table I: please indicate that these are results from synthetic observations (it is explained in the body of the manuscript, but not in the caption). Also, the three angular values are not explained in the caption (nor the table itself), so perhaps the caption could add a sentence to explain that 0.00 deg, 0.01 deg and 0.05 deg refer to the standard deviation of the added noise (added to the AoA or to the relative phases? Please clarify, too).

The caption of Table I has been updated to clarify that these are synthetic retrievals and that the angular values refer to the standard deviation added to the observed AoA:  "The RMS values for the initial refractivity profile and the synthetic retrieved refractivity profile (relative to the reference radiosonde profile).  The three angular values refer to the standard deviation of the added noise to the observed AoA"

Last paragraph of section 3.1, starting at line 258: are these results shown in some of the figures or tables? I could not find them. If they are not shown but only explained in the text, perhaps add at the end 'not shown in the figures' or similar indication.

The following has been added: "The figures for the synthetic refractivity retrievals using the 15 December and 18 July 2022 refractivity profiles are not shown."

Line 276-277 and black points in Figure 10: it is not clear what the black points are. One would think they are the measured AoA (using the interferometric technique), but then they would be noisy, too (while they are absolutely smooth). So I assume they are the line-of-sight AoA, that is, the angle produced assuming the straight line between the interferometer location and the aircraft location. Could this be clarified? Perhaps a set of bullets could summarize all the AoA involved in the analysis, and then relate to these bullets when showing or discussing about one or the other. For example AoA_interf: (measured by interferometer); AoA_LOS (for line-of-sight); AoA_mod0 (result of the synthetic observations modelled based on the initial refractivity profile); AoA_modR (result of the synthetic observations modelled based on the final retrieved refractivity profile).

A new set of bullet points has been added to clarify the variables analysed:

- AoA_LoS: The line-of-sight (LoS) AoA determined from the reported position of the aircraft;

- AoA_init_model: The synthetic LoS AoA determined from the end position of the ray traced through the initial refractivity profile;
- AoA_ret_model: The synthetic LoS AoA determined from the end position of the ray traced through the retrieved refractivity profile;

Line 296: 'and and' (twice, typo)

This has been corrected.

Caption Figure 10 (and/or text around Line 274): please add the time window of this 60 minutes batch. Is it centred at 12:00? or from 11:00 to 12:00? Perhaps part of the disagreement in Figure 11, for time 0-15 min and 15-30 min is due to time variations (Fig 11 shows good agreement for the last two quarter hours, so I wonder if these were closer to the radiosonde).

A statement has been added that the retrieval windows were from 08:36 to 09:36 UTC (the first hour of observations).

Discussion starting at Line 322: how easy it would be to deduce a 'bending' and 'impact' observables as in radio occultation? If this were easy from these measurements, then I believe the 2D forward operators developed for radio occultations and used for their assimilation into NWP could be adapted to this technique. On the one hand, the formulation for bending-impact of reflected signals is already in place, while the geometry in the interferometer is half of the reflected signal in radio occultation. I believe the forward operator for this technique is half (0.5*) Eq.3 in Aparicio et al., 2018 (doi: 10.5194/amt-11-1883-2018), that is, without the 2 factor in both terms. It should not be difficult to adapt the 2D operators to such equation.

This was a very interesting analogy to explore. An adapted geometry of the radio occultation technique was explored in Lewis et al 2023(b). The challenge behind this approach is the unknown emission angle of the radio transmission from the aircraft. This prevents the bending angle being measured directly between the receiver and the aircraft. It was assumed that an estimate could be determined from the model state (since the emission angle can be computed as arcsin(a / nr) where a is the impact parameter, n and r are the refractive index and radial distance of the aircraft respectively), where n is determined from the model. This has not been tested yet so it is not known what impact this would have.

Section 4.3 Multipath: the multipath that comes from nearby constant features (topography, buildings, etc), keeps a constant pattern. Therefore, collecting data for several days, from as many aircraft as possible in as many AoA and Azimuth angles as possible would results in a constant pattern for the oscillations (e.g., after some de-trending to keep only the multipath effect). Once this is well characterized, it can be corrected. See a similar example done in a cliff-based campaign with GPS radio occultations, very much affected by multipath (Fig 3 in doi: 10.5194/acp-16-635-2016).

This was a very useful insight and an additional statement has been added to the manuscript:

"The multipath due to foreground reflections could also be mitigated using a long time series of observations. The impact of reflections from a static foreground would be a systematic perturbation to the observed phase difference. With enough observations from aircraft traversing at a range of elevations and distances the constant multipath interference could be well characterised after accounting for atmospheric variability. The non-constant variability, such as atmospheric refractivity changes, could be de-trended out with sufficient observations. A similar technique is used in ground-based GNSS radio occultation, where Padulles et al (2016) describes how time series of the observations can be used to remove a significant contribution of the multipath effect on the uncertainty in the measurements of the atmospheric polarimetric effects on observables."

More recent experiments have implemented a calibration table of the multipath interference as a function of AoA and azimuth that uses the same idea as above.

Instrumental uncertainties: have you considered the effect of different subsystems that link to only one of the two ends (top/bottom) of the interferometer, such as length and properties of the cables, different RF elements, antenna phase patterns? All these aspects might introduce uncertainties and should be carefully calibrated.

Each array was constructed from identical off-the-shelf components (antennas, attenuators, cables). Some aspects of the design: sub-array cable lengths, antenna placement were dependent on human interaction. Ideally, the individual gains and beam patterns of the antennas would be precisely calibrated but this has not yet been implemented.

*Additional changes*

While making changes to some figures using the suggested comments, the corresponding author discovered a couple of mistakes in the original preprint we would like to correct before publication:

- Figure 5: In the caption, the description for the ray was an AoA of 0.11 deg., an altitude of 11.2 km and a distance of 409.5 km. This should read instead to be an observed AoA of 0.2 deg., an altitude of 8.8 km and distance of 350 km to correctly correspond to the shown figure.

- Figure 8 and Table 1: The synthetic retrievals were generated using an earlier (incorrect) estimate receiver height. After running the synthetic retrievals with the correct receiver height, the plot looks almost identical, although the values in Table 1 are changed slightly as follows (new values and original values in parentheses):
    - 22 Sep 2023: 8.21 (8.64)  0.76 (0.96) 1.42 (1.51) 3.11 (3.59)
    - 18 Jul 2022: 9.63 (9.77) 1.89 (1.81) 3.42 (3.44) 7.09 (8.39)
    - 15 Dec 2022 4.84 (5.10) 0.70 (0.54) 0.88 (1.03) 1.18 (1.73)

Figure 12: The plots shown were computed with the previously mentioned incorrect receiver height. Using the correct receiver height changes the plots slightly, although the conclusions are unchanged.

On line 317, it was stated that no significant frontal activity was present that could

have resulted in the observation noise.  This was incorrect, there was in fact some frontal activity that could have contributed to the observed noise in the retrievals.

The corresponding author would like to apologise for not catching these errors in the earlier preprint.

**Response to Referee 2:**

1 (a) In line 213 of the manuscript a sentence starts saying that "Approximately 263,500 refracted ADS-B transmissions were received between an AoA of 0.0° to 2.0° over an ~ 80-minute period between 08:36 and 09:58 UTC on the 22 September 2023. The synthetic and real retrievals were obtained using the central 10° azimuthal sector of received transmissions". A little later, starting in line 219, it is said that "The adjoint inversion algorithm was tested on synthetic data generated with ADS-B data recorded using the prototype interferometer at Clee Hill, Shropshire. A subset of 5000 ADS-B transmissions were chosen randomly from within the central 10° azimuthal sector". But this subset comes from the whole set of observations (if not, this must be clarified), so I don't understand what the authors mean by "synthetic data".

Thank you for pointing this out, we apologise for the confusion.  The observations throughout the paper were from the central 10 deg. azimuthal sector.  For the synthetic retrievals, 5000 observed AoAs and reported ADS-B positions were used from this central sector.  The following has been added to the manuscript to clarify the process of generating synthetic observations:

"The observations used in the analysis were restricted to the central 10 deg. azimuthal sector of the entire observation dataset.  A subset of 5000 ADS-B transmissions were chosen randomly from within the central azimuthal sector to use in the synthetic refractivity retrieval experiments.   Radiosonde data from the nearby Watnall station in Nottinghamshire (position shown in Fig. 7a) was used to generate a refractivity profile to simulate the refraction of the subset of ADS-B transmissions. Three radiosonde soundings were used to generate the synthetic observations - 12:00 UTC 22 September 2023 (see Fig. 2) 12:00 UTC 18 July 2022 (during an intense heatwave) and 12:00 UTC 15 December 2022 (during a cold spell) (see Fig. 9).  The synthetic observations were generated by tracing the 5000 rays out from the receiver to the reported distances of each of the ADS-B transmissions using the refractivity profiles determined using the radiosonde data.  The interferometrically-measured observed AoAs were used as the initial directions of the rays and the height of the ray endpoint was used as the "true" height of the synthetic ray origin (i.e. equivalent to the height of a synthetic aircraft).  The method is equivalent to that shown in Fig. 3, except the heights of the synthetic aircraft are given by $h_f$  The heights of the synthetic aircraft are treated as the target heights $h_t$ in the synthetic retrievals."

1 (b) Starting in line 227, it is said "Three radiosonde soundings were used to generate the synthetic observations - 12:00 UTC 22 September 2023 (see Fig. 2), 12:00 UTC 18 July 2022 (during an intense heatwave) and 12:00 UTC 15 December 2022 (during a cold spell) (see Fig 8). The simulated samples of refracted ADS-B

transmissions were injected with varying degrees of noise to model the impact of experimental uncertainties on the retrieved refractivity profiles". How are these samples simulated? Are they the same as the 5000 ADS-B transmissions mentioned before? But these were, as far as I understand, real observations, not simulations. This question also applies to table 1: it looks like the authors assume that the simulated AoAs do not have noise, but then, again, I don't see how the simulations are performed. I think this deserves further explanation. A similar comment applies to the conclusions, where it is said that "Three AoA measurement noise cases were simulated" (lines 373-374), but in line 375 it is said that the noise was "injected into the observed AoA measurements".

We apologise for the confusion.  The real ADS-B transmissions and observed AoAs are used to generate synthetic observations using the process described in the response to 1 (a).  The impact of noise in the observed AoA measurement was simulated by adding a random perturbation to each of the 5000 AoAs from a normal distribution with varying standard deviations.

1 (c) It's a little distracting that the refractivity profiles derived from the radiosoundings on 18 July 2002 and on 15 December 2022, shown in figure 8, are discussed after the results in fig. 9. In addition, unless I'm missing something, I don't see a quantitative support to the assertion in lines 261-262, referred to the profile of fig 8a, that "However, in the presence of measurement noise exceeding a standard deviation of 0.01° the complex structure cannot be resolved". What's the contribution of the refractivity profiles in fig. 8 to the method assessment?

Thank you for this helpful suggestion.  Figure 8 has been moved to the end of the section discussing the synthetic refractivity retrievals.  The wording has been made more specific to refer to the inability to capture positive refractivity gradients in the presence of significant observational noise: "However, in the presence of measurement noise exceeding a standard deviation of 0.01 deg. the positive refractivity gradient cannot be resolved."

1 (d) How are the RMS errors in table 1 computed? I miss an equation for that.

We apologise for missing this.  An equation has been added showing the computation of the RMS.

1 (e) In figure 9a, the retrievals of the synthetic refractivity profile (whatever it means) for the two cases of noise in the AoA (note by the way that the units (°) are missing in the standard deviation of the angles in the figure legend), do not seem to be affected by that noise. Shouldn't there be error bars or shaded areas around those retrievals? Likewise, what do the shaded areas in fig. 9a and b represent? Do they mark the limits of the refractivity profiles retrieved with ±5 m and with ±10 m error in the receiver position?

We apologise for this error.  The units have been added to the plot.  Error bars have been added as the range in retrieved profiles for retrievals using twenty different random normally-distributed observed AoA noise cases.  The shaded areas represent the limits of the refractivity profiles retrieved using ±5 m and with ±10 m error in the receiver position (this has been added to the figure caption).

2 (a) In line 275 it is said "Figure 10 shows the reported AoA as a function of horizontal angle". I suppose that the horizontal angle is the elevation angle an aircraft should be seen from the receiver were not for refracting effects; in any case, the meaning of "horizontal angle" should be explained. Perhaps that could be illustrated in fig. 3, which could also be used to illustrate the "local elevation angle", in a similar way as do several figures in reference Zeng et al. 2014, in Lewis et al. 2023a od in Lewis et al. 2023b (sorry for jumping from a section to another).

The corresponding author apologises for the confusing terminology.  In Figure 10 the "horizontal angle" actually refers to the "azimuthal angle" (although 0 deg. is not due north, but in the direction of the central beam of the receiver).  This has been clarified in the caption of Fig. 10.

2 (b) Without any explanation, the statement in lines 276-277, "The black points (obs) indicate the reported AoA determined from the ADS-B positional data (i.e. the "true" reported flight paths)" seems strange at first thought, since the AoA is not reported (at least directly) by the ADS-B system, but given by interferometer. Perhaps the authors mean that the phase ambiguity resulting from a large baseline/wavelength ratio is resolved using the height reported by the ADS-B, as explained in Lewis et al. 2023a. If this is the case, I think that a brief explanation would be helpful.

We apologise for the confusing terminology.  A new set of bullet points has been added to clarify the variables analysed:

- AoA_LoS: The line-of-sight (LoS) AoA determined from the reported position of the aircraft;
- AoA_init_model: The synthetic LoS AoA determined from the end position of the ray traced through the initial refractivity profile;
- AoA_ret_model: The synthetic LoS AoA determined from the end position of the ray traced through the retrieved refractivity profile;

The LoS AoA is now referred throughout, being the angle above the horizontal of a straight line drawn between the receiver and the aircraft (for the reported ADS-B position) or the ray end position (for the simulated cases).

2 (c) Line 277: How are the "retrieved flight paths using the initial (prior) exponential profile" obtained? I understand, from the explanations in section 2.3, that the retrieved ray is essentially obtained by propagating from the receiver to the aircraft horizontal position and adjusting iteratively the refractive indices of the layers the atmosphere is divided into until the quadratic difference between the ray endpoint height and the reported aircraft height is minimized, but how is the AoA determined from just the initial refractivity profile? Is it done as suggested in section 5.1 of Lewis et al. 2023a? If this is the case, I think a brief explanation should be included in the present paper or, at least, an indication to the reader to consult that reference close to the mention of the "retrieved flight paths using the initial (prior) exponential profile".

We apologise for the confusing terminology.  The AoA referred to here is the line-of-sight (LoS) AoA.  This is determined by tracing a ray out from the receiver to the

aircraft horizontal position and computing the angle above the horizontal of the straight line joining the receiver and end point of the ray (this is equivalent to the observed AoA if no refraction occurred). An additional statement has been added referring to Lewis et al 2023a.

2 (d) Line 285: Is the "observed AoA measurement" the same as the "true reported AoA"? Is the use of "reported" in "retrieved reported AoA" necessary? Why? Related to this, is the "retrieved modelled reported AoA" mentioned in the caption of fig. 12 the same as the "retrieved reported AoA" mentioned in line 285? Maybe a set of definitions of the different AoAs considered would be helpful, even if these definitions are found in Lewis et al 2023a, in which case this should emphasized.

We apologise for the confusing terminology. The "observed AoA" refers to the AoA determined using the interferometer. The "true reported AoA" referred to the LoS AoA determined using the ADS-B positional information. These have been updated in the bullet point list described in response to 2(b), with the true reported AoA refereed to as AoA_LoS.

2 (e) Is it possible to assign uncertainties (error bars) to the retrieved refractivity profiles shown in fig. 11?

At this stage it is very difficult to assign uncertainties to the retrieved refractivity profile. The retrieved profile is generated using thousands of individual ADS-B signals, where the uncertainty in the observed AoA for each is unknown. We attempt to quantify the uncertainty using the LoS AoA of the end point of the ray through the retrieved refractivity profile and the LoS AoA determined using the ADS-B reported position, where the large standard deviation in Figure 13 demonstrates the uncertainty in the retrieval in terms of measurable quantities. A better estimation of the actual uncertainty in the refractivity will likely require sensitivity/assimilation experiments using a numerical weather prediction model, with initial experiments being started using the Data Assimilation Research Testbed (DART).

A final, not fundamental, remark on something that can mislead a reader: were not for the paper title, reading only the abstract and especially the introduction, one could think that the paper is going to deal with humidity profiles. I understand that the refractivity profile provides humidity information, but somehow the introduction should make clear that the paper deals with refractivity, not humidity, profiles

We apologise for the lack of clarityThe following has been added on line 47 of the original manuscript:

"High resolution direct/indirect measurements of humidity are challenging to obtain. However, the high spatial and temporal variability of water vapour in the lower atmosphere results in significant changes in the optical properties of the atmosphere. A valuable source of indirect humidity information for use in NWP is provided through observations of refractivity…"

*Additional changes*

While making changes to some figures using the suggested comments, the corresponding author discovered a couple of mistakes in the original preprint we would like to correct before publication:

- Figure 5: In the caption, the description for the ray was an AoA of 0.11 deg., an altitude of 11.2 km and a distance of 409.5 km. This should read instead to be an observed AoA of 0.2 deg., an altitude of 8.8 km and distance of 350 km to correctly correspond to the shown figure.

- Figure 8 and Table 1: The synthetic retrievals were generated using an earlier (incorrect) estimate receiver height. After running the synthetic retrievals with the correct receiver height, the plot looks almost identical, although the values in Table 1 are changed slightly as follows (new values and original values in parentheses):
    - 22 Sep 2023: 8.21 (8.64)  0.76 (0.96) 1.42 (1.51) 3.11 (3.59)
    - 18 Jul 2022: 9.63 (9.77) 1.89 (1.81) 3.42 (3.44) 7.09 (8.39)
    - 15 Dec 2022 4.84 (5.10) 0.70 (0.54) 0.88 (1.03) 1.18 (1.73)

Figure 12: The plots shown were computed with the previously mentioned incorrect receiver height. Using the correct receiver height changes the plots slightly, although the conclusions are unchanged.

On line 317, it was stated that no significant frontal activity was present that could have resulted in the observation noise. This was incorrect, there was in fact some frontal activity that could have contributed to the observed noise in the retrievals.

The corresponding author would like to apologise for not catching these errors in the earlier preprint.

---

## Author Response (AR2)

**Response to Referee 2:**

We would like to thank the referee for their insightful and very helpful comments that have improved the clarity of this manuscripts. We would like to address their latest points below.

1) The so-called synthetic observations outline in lines 245-250 of the revised manuscript ("The synthetic observations were generated using 5000 rays, etc.") need in my opinion some clarifications in the text:

    a) Does that mean that 5000 heights of synthetic aircrafts were generated?

Yes. This has been clarified in the main text as "The synthetic observations (heights of synthetic aircraft) were generated using 5000 rays…".

    b) Starting from the prior given by Eq. (23), were 5000 synthetic refractivity profiles calculated? In that case, how were the profiles of fig. 8a obtained? Perhaps averaging the 5000 profiles? Was the 0.00° RMS profile obtained from the 5000 interferometrically-measured observed AoAs. Were the 0.01° and the 0.05° RMS profiles obtained by perturbing the 5000 interferometrically-measured observed AoAs according to the corresponding normal distributions?

The following has been added on line 280 for clarification:

"The 0.00 deg. RMSE profile was obtained from the 5000 interferometrically-measured, unperturbed observed AoAs. The 0.01 deg. and the 0.05 deg. RMSE profiles were obtained by perturbing the 5000 interferometrically-measured observed AoAs according to the corresponding normal distributions."

    c) I don't understand the final sentence in the caption of fig. 8: "The error bars show the range in retrieved refractivity profiles using twenty different observed AoA noise distributions." What is understood by "noise distribution" in this context? Do the authors mean "noise realizations" instead of "noise distributions"? Wouldn't it be more rigorous to compute (and represent) the standard deviation of the noisy profiles?

Yes, this should be noise realisations. The fractional error standard deviation is included in Fig 8b for the twenty noisy retrievals. The following has been added to line 284, describing the bias increasing as the AoA noise increases:

"The 0.05 deg. RMSE profile shows a lower refractivity error standard deviation at higher altitudes than the 0.01 deg. RMSE profile. The increased standard deviation in the AoA measurement error results in a greater number of AoAs being perturbed below 0 deg. for the 0.05 deg. RMSE case. Since the retrievals can currently only be performed using AoAs ≥ 0 deg., these are then rejected, resulting in a bias towards perturbing the AoAs towards higher elevation angles. This results in the gradient in the retrieved refractivity profile becoming steeper which, as a

consequence of the clamping of minimum refractivity, results in the 0.05 deg. RMSE profiles clustering together with a lower variance.  However, the larger RMSE in the observed AoA results in a larger bias in the retrieved refractivity profile."

2) I still have concerns on the message figures 9a and 9b are meant to convey. On the one hand, the synthetic retrieved refractivity profiles of fig. 8 fit quite well the radiosonde-derived profile and one can think that this is because, despite they have been calculated on the synthetic aircraft positions, these positions were generated in turn using the radiosonde refractivity profile. On the other hand, no retrieved synthetic profiles are shown in fig. 9, just the radiosonde-derived profiles ("The figures for the synthetic refractivity retrievals using the 15 December and 18 July 2022 refractivity profiles are not shown") and one has to make do with the assertions that, even injecting noise, there is improvement with respect the initial guess profile, but that, under noise with 0.01° (or greater) standard deviation, the positive refractivity gradient in fig. 9a cannot be resolved. To this point, I don't see the information added by the figures beyond the fact that negative refractivity gradients can happen. In addition, if I'm not wrong, the observed AoAs from 22 September 2023 were also used (at least there is no mention to other AoA observations) to generate synthetic retrievals in these two different days, several months earlier. To what extent is this meaningful? Is it sensible to use observed AoAs obtained in some conditions of refractivity profile to generate synthetic aircraft positions in (very) different conditions to test the retrieval algorithm in the latter conditions and trying to obtain valid information? Can it be excluded that the failure to retrieve profiles with negative refractivity gradients in the presence of injected noise is caused by this mismatch between the observations obtained under a given profiles and the synthetic aircraft positions being calculated under a potentially very different profile? All this in turn prompts a general question about the purpose of the synthetic retrievals: are they just intended to test the algorithm and its robustness against noise? Does this have an impact on the "real" retrievals of fig. 11?

We thank for the referee for their helpful suggestions.  We have removed Fig. 9, showing the radiosonde profiles (but no retrievals) and instead opted to just describe the results in the main text and in Table 1.  The synthetic retrievals for the 18 July and 15 Dec were obtained using an identical process to that used to generate the 22 Sep 2023 data.  The respective profiles were used to obtain the "true" synthetic aircraft heights, then noise was added to the AoAs and the original profiles were retrieved using these noisy AoAs and synthetic aircraft heights.  A description has been added in the manuscript in the last paragraph of section 3.1:

"The synthetic refractivity retrievals were then repeated for two other radiosonde profiles - the 12:00 UTC Watnall soundings for the 18 July 2022 and 15 December 2022 respectively.  The synthetic retrievals were performed using synthetic observations of aircraft heights generated from the two respective refractivity profiles in a process identical to that using the 22 September data."

The following was also added, to summarise the reasoning behind the synthetic retrievals:

"The synthetic refractivity retrievals demonstrate the ability to obtain useful atmospheric data using measurements of refracted ADS-B transmissions in the presence of random noise."

3. Connected to the last question, I think that figure 11, or its context, could be more informative. For example, how many observations were used to generate the profile shown for each time slot? If, as I suppose, several observations were used, how is the shown profile obtained (see question 1b). Could the variability of the profiles generated from observations within a time slot be used to assess the uncertainty of the final profile? Likewise, how many observations have been used in the graphs of figure 12 for each of the time slots. Are they the same used to generate the profiles of fig. 11?

The following statement was added to the first paragraph of section 3.2:

"The four observational periods contained 9700 (08:36-08:51 UTC), 8315 (08:51-09:06 UTC), 8185 (09:06-09:21 UTC) and 7589 (09:21-09:36 UTC) received ADS-B transmissions respectively within the sector."

The number of observations in each observational period was also included in the caption for Fig. 10.

In terms of estimating the refractivity uncertainty in the retrievals, this is currently challenging to estimate.  The retrievals require a suitable distribution in aircraft heights (since it is a tomographic problem), and neglecting certain aircraft heights has a significant impact on the retrieval quality.  Future experiments plan to assimilate the data into numerical weather prediction models (rather than direct retrievals of vertical profiles) using either a variational approach or through the use of ensemble Kalman filters.    The observations for the four periods were the same for Figs 9, 10 and 11 (stated in main text).

4. The main source of uncertainty in the observations is attributed to multipath (lines 259-261: "Since multipath contamination of the observed AoA measurement is thought to constitute the principal source of uncertainty in the retrieved refractivity profile, three observed AoA measurement noise test cases were investigated"). Can these multipath effects be mimicked by normally-distributed noise? In case it is justified they can, what's the amount of noise to be expected for the real retrievals?

This is currently outside the scope of the paper, as the multipath environment is extremely complex.  Future experiments will likely require wave optics modelling to determine the uncertainty arising due to multipath.  Some AoAs are also biased, which is difficult to account for in the retrievals currently.  A statement has been added to the end of section 4.3:

"Future experiments using the prototype ADS-B interferometer will aim to quantify the nature of the observational noise arising due to multipath contamination."

Minor remarks:

1. Is it necessary the redundancy between what is said in lines 227-228 of the revised manuscript ("The synthetic and real retrievals were obtained using the central 10∘ azimuthal sector of received transmissions") and the sentence in lines 234 and 235 (" The observations used in the analysis were restricted to the central 10° azimuthal sector of the entire observation dataset")?

This has been corrected, including just the first statement.

2. I suggest to modify the writing at the beginning of line 245 from "the reported distances of each of the ADS-B transmission" to "the reported distances s_T (see fig. 3) of each of the ADS-B transmission".

This has been added.

3. I would suggest to modify the sentence in the last two lines of page 13 of the revised manuscript, which currently reads "A prior (first-guess) refractivity profile was used to initialise the penalty function, given by", to "A prior (first-guess) refractivity profile, given by

$$N(h)=N_0 \exp(-h/H), \quad (23)$$

was used to initialise the penalty function". This is to make clear that Eq. (23) is the first-guess refractivity profile, not the penalty function.

This has been corrected.

4. Vertical axes of the panels in figs. 11 and 12: the labels should probably be "LoS AoA (deg.)" or "AoA_Los" in view of the new terminology defined in lines 312-316.

This has been corrected.

5. Line 433: "The convergence of the penalty function was impacted negatively and the RMS in the retrieved refractivity profile". There is a noun (probably "error") missing after RMS, which acts as an adjective.

This has been corrected, and throughout has been corrected to RMSE (rather than RMS).